# Warming drives dissolved organic carbon export from pristine alpine soils

Andrew R. Pearson [1,2] ✉, Bethany R. S. Fox [3], John C. Hellstrom [4], Marcus J. Vandergoes[5], Sebastian F. M. Breitenbach [6], Russell N Drysdale [4], Sebastian N. Höpker[1], Christopher T. Wood[1,5], Martin Schiller [7] & Adam Hartland [1,8] ✉

Despite decades of research, the influence of climate on the export of dissolved organic carbon (DOC) from soil remains poorly constrained, adding uncertainty to global carbon models. The limited temporal range of contemporary monitoring data, ongoing climate reorganisation and confounding anthropogenic activities muddy the waters further. Here, we reconstruct DOC leaching over the last ~14,000 years using alpine environmental archives (two speleothems and one lake sediment core) across 4° of latitude from Te Waipounamu/South Island of Aotearoa New Zealand. We selected broadly comparable palaeoenvironmental archives in mountainous catchments, free of anthropogenically-induced landscape changes prior to ~1200 C.E. We show that warmer temperatures resulted in increased allochthonous DOC export through the Holocene, most notably during the Holocene Climatic Optimum (HCO), which was some 1.5–2.5 °C warmer than the late pre-industrial period— then decreased during the cooler mid-Holocene. We propose that temperature exerted the key control on the observed doubling to tripling of soil DOC export during the HCO, presumably via temperature-mediated changes in vegetative soil C inputs and microbial degradation rates. Future warming may accelerate DOC export from mountainous catchments, with implications for the global carbon cycle and water quality.

Soils represent the largest terrestrial pool of carbon and store more carbon than terrestrial vegetation and the atmosphere combined[1,2]. Soil organic carbon (SOC) stocks reflect a balance between numerous processes influenced by climate[3], including net primary production[4], microbial priming and decomposition rates[5], and SOC solubility[6–8]. The release of dissolved organic carbon (DOC) from soils represents an important but uncertain flux within the global carbon cycle that remains poorly constrained, or not represented at all, in global carbon budgets[9–11].

Most DOC in freshwaters derives from plant tissue (following biological, physical and chemical processing in soil[12]). DOC export is thus directly linked to carbon storage in catchment soils, constituting the least constrained aspect of this mass balance (Eq. (1))[6]:

$$SOC = C_{PP} - C_R - C_{DOC} \qquad (1)$$

[1]Environmental Research Institute, School of Science, Faculty of Science and Engineering, University of Waikato, Kirikiriroa Hamilton, Waikato, Aotearoa, New Zealand. [2]Institute of Environmental Science and Research (ESR), Ōtautahi Christchurch, Aotearoa, New Zealand. [3]Department of Biological and Geographical Sciences, University of Huddersfield, Huddersfield, UK. [4]School of Geography, Earth and Atmospheric Sciences, University of Melbourne, Melbourne, VIC, Australia. [5]GNS Science, Te Awa Kairangi ki Tai Lower Hutt, Aotearoa, New Zealand. [6]Department of Geography and Environmental Sciences, Northumbria University, Newcastle upon Tyne, UK. [7]Centre for Star and Planet Formation, Globe Institute, University of Copenhagen, Copenhagen, Denmark. [8]Lincoln Agritech Ltd, Ruakura, Kirikiriroa Hamilton, Waikato, Aotearoa, New Zealand. ✉e-mail: Andrew.Pearson@esr.cri.nz; Adam.Hartland@waikato.ac.nz

where $C_{PP}$ is the carbon fraction fixed by primary production, $C_R$ is the respired carbon fraction (as $CO_2$), and $C_{DOC}$ is the carbon fraction lost via DOC solubilisation and export.

Soluble carbon fluxes from soil are expected to form a 'climate feedback' as the world continues to warm[5]. Yet, the response of DOC export to rising global temperatures and changes in hydroclimate also has implications for freshwater and marine ecosystems. DOC increases in aquatic systems have significant impacts, including eutrophication[7], reduced surface water clarity[9,13], and enhanced contaminant transport[8], as well as influencing groundwater pH[14] and fuelling redox transformations in the subsurface[15–17]. Generally, aquatic DOC concentrations are positively related to SOC stocks in developed soils, and vegetation cover[18], but can also be influenced by the presence of peat, which can export up to 10 times more DOC than forest soils[19].

Monitoring programmes, like the UK Acid Waters Monitoring Network (AWMN), provide multi-decadal records of aquatic DOC concentrations allowing us to evaluate impacts on receiving environments[20]. But, the analysis of climate-DOC feedbacks is confounded by human perturbation of landscapes (e.g., agricultural land-use and urbanisation)[21] and atmospheric chemistry (e.g., increased N and S deposition associated with fossil fuel combustion) over this time interval[22–24]. While some studies report links between aquatic DOC concentration and recovery from soil acidification on continental scales[14–16], others show positive links between atmospheric temperature and DOC export[25–27]. For example, an assessment of 315 records (≥10 years) of riverine DOC across Great Britain showed that increasing concentrations were correlated with increasing air temperature and atmospheric $CO_2$[28]. Additionally, anthropogenically impacted soils may respond differently to projected warming than soils in undisturbed areas. For example, a recent review of carbon fluxes from peatlands attributed increased DOC export from undisturbed peatlands to warming air temperature, whilst DOC export from disturbed peatlands was controlled by outlet discharge[29]. Despite DOC changes coinciding with warming temperatures since the mid-to-late twentieth century, the limited temporal (i.e., decadal-scale) coverage of monitoring and the confounding impact of anthropogenic activity mean that the long-term response of soil DOC export to atmospheric warming remains poorly constrained[4,30] (including in catchments that are relatively pristine[29]). Further, although individual processes may respond rapidly to climatic change, the whole ecosystem response may take decades or even centuries to manifest[31]. Thus, the influence of climate on soil DOC export is difficult to assess via conventional approaches.

Palaeo-environmental archives record catchment or regional-scale changes over millennia and provide longer-term perspectives compared to instrumental monitoring data[32]. Environmental archives from Aotearoa New Zealand (referred to hereafter as Aotearoa) are particularly valuable for assessing climatic influence on DOC feedbacks because of the absence of permanent human residence until ~800 years ago[33,34]. This relatively short human impact period allows us to consider the influence of climate on DOC export without the confounding influence of human impacts prior to ~1200 C.E. Furthermore, some periods of Aotearoa's climatic past can serve as analogues for future warming. For example, during the Holocene Climatic Optimum (HCO; ~12.5−9.5 kyrs ago)[35], the air temperature was 1.5–2.5 °C warmer than during the immediate pre-industrial period, with diminished seasonality[36–38]. Climate models project a reduced seasonal temperature contrast and a rise in mean annual temperature of between +0.7 °C (RCP2.6 scenario) and +3.7 °C (RCP8.5) by 2100 (relative to 1986–2005) for Aotearoa[35]. Hence, Aotearoa's HCO represents an analogue of projected climate change for the mid-century (2055) under RCP8.5 or end of century (2100) under RCP4.5[35].

Our aim was to assess the long-term (the last ~14 kyr) influence of climate on DOC export from sub-alpine soils using speleothem (secondary cave carbonate deposits) and lake sediment archives. Although

most speleothem research focuses on inorganic climate proxies, speleothems can also incorporate organic material, including lignin phenols, lipid biomarkers[39–41] and DOC[42]. Soils overlying caves leach DOC, which is transported through the vadose zone before being coprecipitated into speleothems and preserved over millennia[8,43,44]. Speleothems therefore offer unique archives of soil DOC dynamics[25], owing to their position in the unsaturated zone and their ability to capture and preserve soil-derived fluorescent organic matter from infiltrating water[45–51].

Here, we use 3D excitation-emission matrix (EEM) fluorescence to measure DOC concentrations in speleothems, enabling reconstruction of soil export at two sites at different latitudes on Te Waipounamu South Island (referred to hereafter as Te Waipounamu): Hodge Creek Cave (41°S) and Dave's Cave (45°S) (Fig. 1). EEM fluorescence spectroscopy[52–54] is an established technique for DOC monitoring and characterisation which is linear with DOC concentration across the range observed in groundwater[14]. In addition, cave carbonates reliably record aqueous DOC concentrations during calcite precipitation[42], making them uniquely suitable for DOC reconstructions. To verify our speleothem results, we compared the speleothem DOC records to a sediment-based reconstruction from Adelaide Tarn, an alpine lake ~32 km from Hodge Creek Cave and located at a similar altitude. DOC was extracted from Adelaide Tarn sediments using a water extraction method commonly used in soil analysis[55] and analysed using EEM fluorescence[56]. Further, we applied a partial-least squares regression model to infer total organic carbon (TOC) concentrations through time via Fourier-transform infrared spectroscopy (FTIRS) of the sediment aliquots. FTIRS-TOC has previously been applied to reconstruct TOC concentrations in European and Aotearoa lakes[56–60]. In Aotearoa, there are few reconstructions of hydrological change, particularly in alpine areas. Thus, we also include elemental (Mg/Ca and Sr/Ca) and isotopic ($\delta^{44}Ca$) speleothem data, which are proxies for prior calcite precipitation (PCP) and are increasingly applied to indicate hydrological change[61,62]. Our interpretations of environmental changes are further supported by flowstone oxygen ($\delta^{18}O$) and carbon ($\delta^{13}C$) isotope reconstructions.

## Results and discussion
### Study region and site locations
Alpine and sub-alpine areas were preferred because of their sensitivity to change: warming rates and landscape response times are amplified at elevation, and thus high-altitude archives represent sentinels of global environmental change[63,64]. This study uses two flowstone cores and one lake sediment core, each of which accumulated over the last ~14,000 years (as determined by uranium–thorium (U–Th) disequilibrium dating of the speleothems and radiocarbon dating of Adelaide Tarn lake sediment[65] (Supplementary Information)). During precipitation from solution, speleothems incorporate soil-derived DOC[39,40,43] linearly across the relevant DOC concentration range (0–15 mg l$^{-1}$)[42]. Lake sediments are well-established archives of catchment-derived DOC and have been used for reconstructions of DOC concentrations[32].

Aotearoa is an archipelago situated in the mid-latitudes of the southwest Pacific Ocean (Fig. 1). Te Waipounamu is bisected by the Southern Alps (maximum altitude >3700 m), which span the length of the landmass (Fig. 1a). Te Waipounamu has a temperate maritime climate governed by the prevailing westerly circulation and orographic rainfall concentrated over the Southern Alps[66], and is influenced by both subtropical and subantarctic waters, the prominence of each likely having varied through the Holocene[37,67].

We selected high-altitude study areas that have been minimally disturbed by human activity. Each site is sub-alpine (Hodge Creek Cave 940 m above sea level (a.s.l.), Dave's Cave 1450 m a.s.l., and Adelaide Tarn 1270 m a.s.l.) (Fig. 1) and located on the western side of the Main Divide. Adelaide Tarn and Hodge Creek Cave are co-located in

Kahurangi National Park, which has an essentially maritime climate[66], dominated by prevailing westerly fronts from the Tasman Sea, ~50 km to the west. Northwestern Te Waipounamu is also influenced by the proximity of Cook Strait, which divides Aotearoa's main islands. In Kahurangi National Park, north-westerly winds are funnelled between the islands, displacing the south-westerly winds which are dominant to the north and south[68]. Paleoclimate records from this region show correspondence to Tasman Sea climate, and are therefore regional in character[69].

Hodge Creek Cave (41°S, 172°E) is located on Mount Arthur in the Arthur Range of the Southern Alps (Fig. 1a). The cave developed in heavily weathered Oligocene limestone and is overlain by dense stands of native silver beech (*Nothofagus menziesii*) with ferns and mosses in the lower canopy and a dark, organic-rich soil which fills deep grykes, promoting gleization. Hodge Creek Cave is positioned below the present treeline (1200–1300 m a.s.l.), which corresponds to a coldest-month mean temperature of ~0 °C[70,71]. Several previous studies have

demonstrated the sensitivity of Mount Arthur's speleothems to climatic and vegetative changes (from several different caves positioned at various elevations)[68–70], and Hodge Creek Cave was therefore expected to exhibit similar sensitivity.

Adelaide Tarn (41°S, 172°E) is a small (0.06 km²), shallow (maximum depth 7.6 m) lake, that sits within a glacial cirque (3.8 km²) in the Douglas Range[65], positioned around ~32 km from Hodge Creek Cave (Fig. 1). Notably, since the lake's formation 16 kyrs ago, regional vegetation dynamics were sensitive to temperature (as demonstrated from palynological analysis of Adelaide Tarn's sediments)[65]. Further, the presence of tree macro-fossils provide unequivocal evidence for the first arrival of trees in Adelaide Tarn's basin towards the end of the HCO (~9.7 kyrs ago), with the advancement in treeline elevation (despite warming temperatures) delayed by steep slopes and rugged topography[65]. Although a period of prolonged cooling followed the HCO, Adelaide Tarn's basin remained forested until ~2.7 kyrs, when cooling forced the treeline retreat to a lower elevation[65]. Currently,

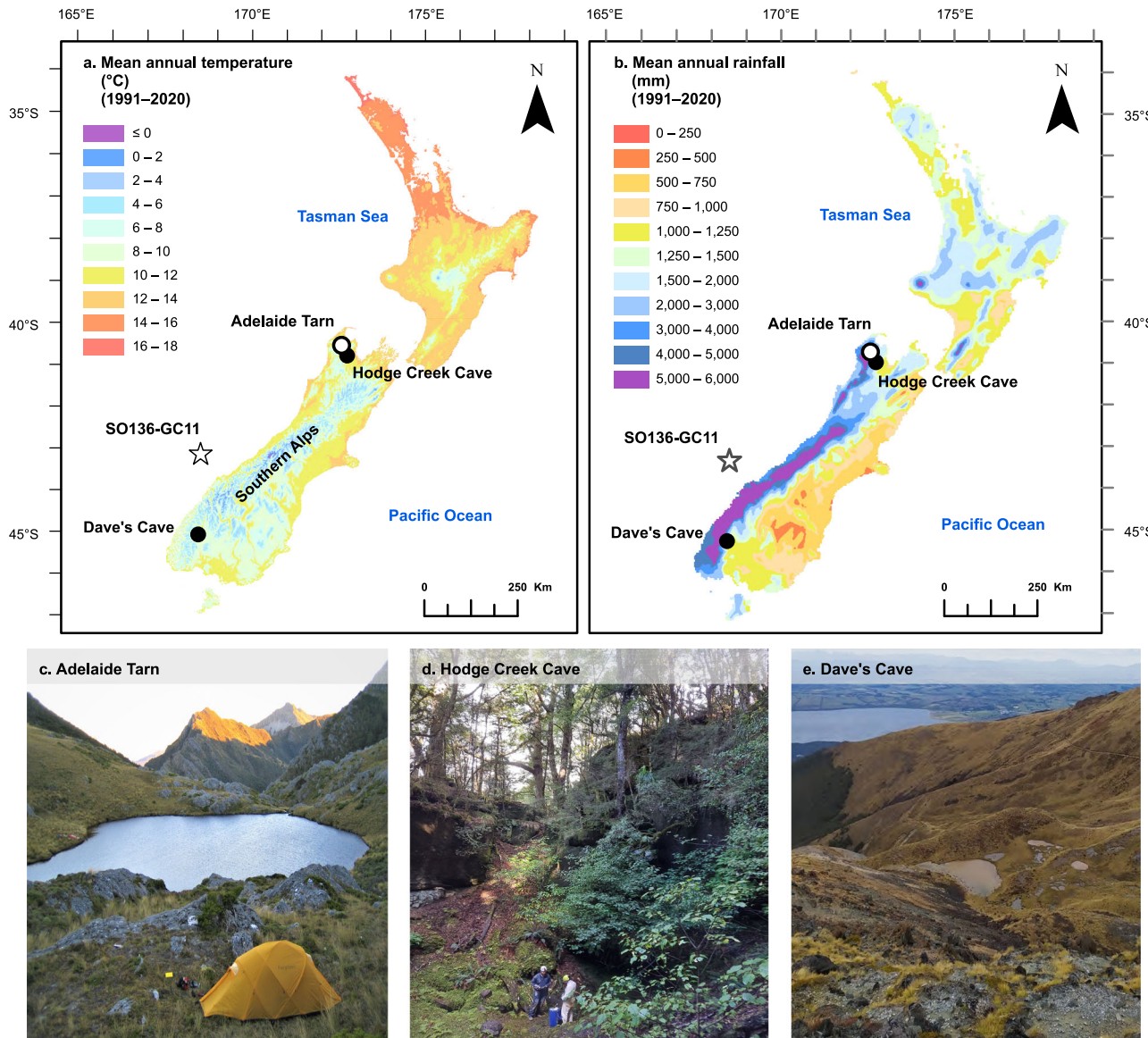

**Fig. 1 | Site locations and settings within Aotearoa. a** Aotearoa mean annual temperature (1991–2020) contours and locations of Adelaide Tarn, Hodge Creek Cave, and Dave's Cave. The alkenone-based sea-surface temperature (SST) reconstruction[38] originates from site SO136-GC11 west of Te Waipounamu. **b** Map of Aotearoa's mean annual rainfall (1991–2020). **c** Adelaide Tarn, situated on the Douglas Range of Kahurangi National Park. **d** Landscape overlying Hodge Creek Cave. **e** Landscape overlying Dave's Cave, which is located above the treeline on Mount Luxmore. Mean annual temperature and rainfall data courtesy of the National Institute of Water and Atmospheric Research (NIWA).

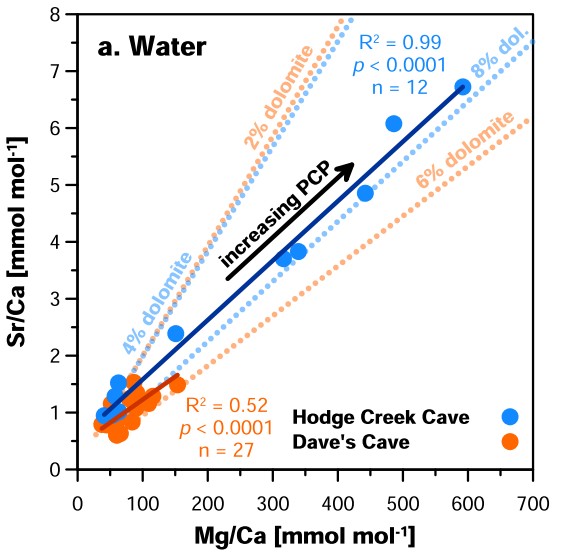

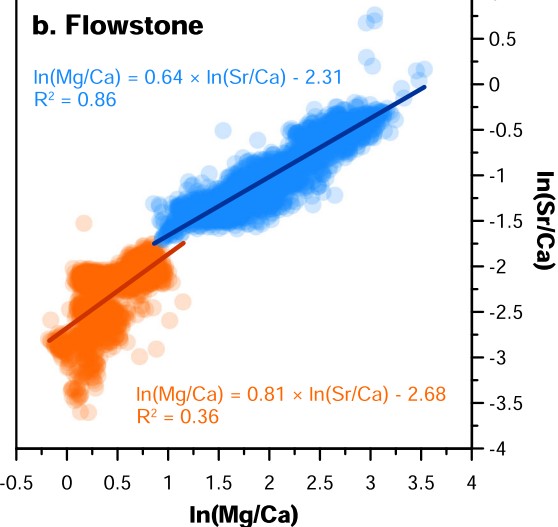

**Fig. 2 | Mg/Ca and Sr/Ca in dripwaters and flowstone samples. a** Atomic ratios of Mg and Sr to Ca in dripwater samples from Hodge Creek Cave (blue dots) and Dave's Cave (orange dots). Solid coloured lines reflect respective linear fits. Dotted coloured lines indicate theoretical relationships between Mg/Ca and Sr/Ca as a function of PCP for each cave, calculated for a limestone bedrock with small contributions of Mg from dolomite[84]; see 'Methods' for details). **b** Flowstone Mg/Ca and Sr/Ca signatures (expressed as natural logarithms), including all data shown in the timeseries in Fig. 3. Solid lines reflect respective linear best fits. For both speleothems, the slopes of this relationship cohere with a dominant hydrological control due to PCP.

Adelaide Tarn's basin is unforested with tussock grassland of the genus *Chionochloa* predominating, despite being lower than the average treeline (dominated by *Fuscospora cliffortioides* (mountain beech)) elevation in the Douglas Range (~1350 m a.s.l.)[65].

Dave's Cave (45°S, 168°E) resides above the mountain beech treeline on Mount Luxmore, overlooking Lake Te Anau in Fiordland National Park. The vegetation above Dave's Cave (Fig. 1d) consists mainly of alpine shrubs and tussocks, whilst the soil is organic-rich and poorly drained. Air masses from tropical and polar regions can reach Fiordland, where heavy rainfalls and cold showery conditions are common[72].

**Inorganic and prior calcite precipitation proxies in caves**

Given the relative dearth of paleo-hydrologic data from Aotearoa, we developed a multi-proxy dataset based on speleothem PCP proxies[61,62], which are increasingly applied in speleothem paleoclimate studies to indicate hydrologic change, in addition to more conventional oxygen ($\delta^{18}O$) and carbon ($\delta^{13}C$) isotope ratios. PCP proxies respond to the precipitation of calcite along the speleothem flowpath and broadly reflect drier, better-ventilated conditions within the epikarst, karst aquifer, and cave environment. In some speleothems, Mg/Ca and Sr/Ca provide insight on effective infiltration (and therefore effective rainfall) through PCP and other controls, which increase Mg/Ca and Sr/Ca ratios during drier periods[73]. However, results can be confounded by local hydrological controls on water evolution and flow paths[74]. Calcium isotope ratios ($\delta^{44/42}Ca$) have emerged as a proxy for local infiltration[75,76], as the lighter isotope ($^{42}Ca$) is preferentially precipitated during PCP[75,77]. Factors controlling oxygen isotopes ($\delta^{18}O$) are numerous and complex[78], including sensitivity to cave temperature and effective precipitation (amount, or moisture source)[72,79], however more positive values may indicate lower rainfall when the amount effect is active. $\delta^{13}C$ behaviour in cave systems is also highly complex[80], and can be influenced by soil respiration[81], and in-cave processes such as drip rate and degassing[82]. $\delta^{13}C$ variability in high-altitude speleothems has also been attributed to changes in vegetation cover and/or soil thickness[83], including in Mount Arthur flowstones[68]. However, $\delta^{13}C$ can also move positively due to PCP, and covariation with Mg/Ca can suggest PCP influence. Additionally, covariance between $\delta^{13}C$ and $\delta^{18}O$ may represent increased kinetic fractionation, another indicator of relatively dry conditions in the epikarst.

PCP proxies respond to drier conditions because aridity increases the potential for gas exchange along dripwater flow paths. While PCP proxies have been interpreted within quantitative frameworks elsewhere[62], a fully quantitative treatment requires empirical functions between moisture balance and PCP, and further necessitates long-term monitoring datasets that are beyond the scope of this study. However, both Mg/Ca and Sr/Ca in Hodge Creek Cave and Dave's Cave dripwater largely cohere with expected evolutions due to PCP (Fig. 2a)[84]. In the flowstone records, this imprint of PCP is supported by the slopes of ln(Mg/Ca) versus ln(Sr/Ca) (Fig. 2b) of ca. 0.64 and 0.81 for Hodge Creek Cave and Dave's Cave, respectively[74]. Although additional processes (e.g., incongruent calcite dissolution) likely contribute to Mg and Sr signatures (particularly in Dave's Cave), these data support the overall interpretation of karst hydrology as the dominant control.

To buttress our interpretations from trace element records from Hodge Creek Cave, we include published oxygen ($\delta^{18}O$) and carbon ($\delta^{13}C$) isotope records from Exhaleair Cave and Nettlebed Cave[68,69], which (like Hodge Creek Cave) are positioned on Mount Arthur, albeit at lower elevations (685 m and 390 m a.s.l., respectively). From the Dave's Cave flowstone, we present Mg/Ca and Sr/Ca alongside $\delta^{44}Ca$, $\delta^{18}O$, and $\delta^{13}C$ (Fig. 3). Allowing for these caveats, the hydrological proxy data at large show coherent trends and allow qualitative interpretations with which to test the controls on DOC variations presented in Fig. 4. The evolution of hydrological changes is discussed in relation to DOC dynamics in the remainder of the text.

**DOC characteristics in dripwaters and flowstones**

The composition of fluorescent DOC in dripwater and the environmental archives, as identified by PARAFAC models of EEM fluorescence (Fig. 4), was broadly humic in character, with some minor spectral variance from site to site. Analysis of water-extractable organic carbon from Adelaide Tarn sediments revealed both humic-like (peaks C and A (following nomenclature described by Coble et al.[54]) (excitation wavelength 250–260 nm (secondary peak at 280–330 nm); emission wavelength 380–480 nm) and protein-like (peak T (excitation wavelength: 250–280 nm; emission wavelength: 300–340 nm)) PARAFAC fluorescence components. At Hodge Creek Cave, fluorescence analysis

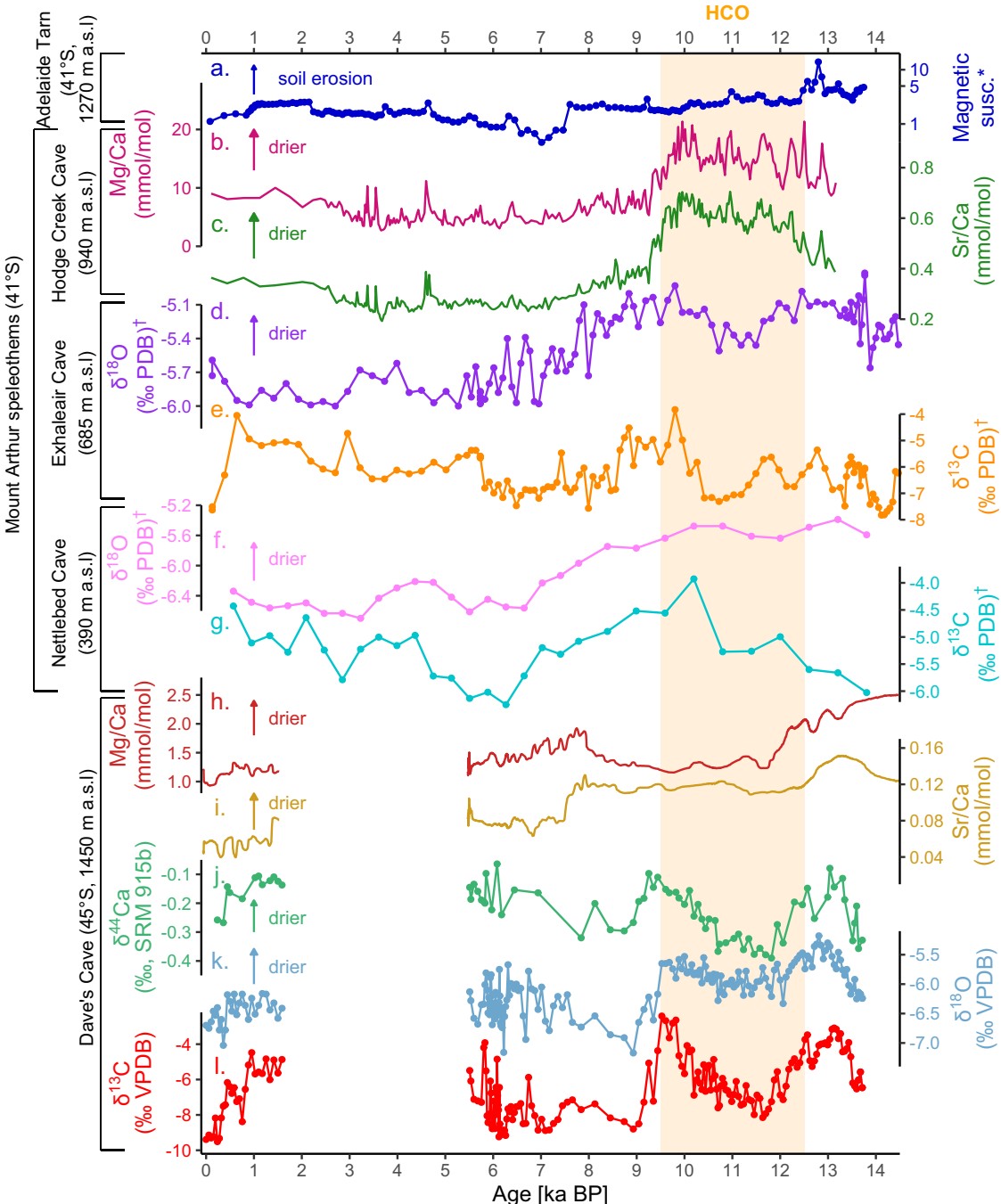

**Fig. 3 | Key local palaeoclimate records from this study in addition to spe-leothem records from Mount Arthur and Mt Luxmore caves. a** Magnetic sus-ceptibility in Adelaide Tarn's sediments (a proxy for soil stability) (*data from Jara et al.[65]); Hodge Creek Cave flowstone (**b**) Mg/Ca and (**c**) Sr/Ca; Exhaleair Cave flowstone (**d**) δ18O and (**e**) δ13C; Nettlebed Cave flowstone (**f**) δ18O and (**g**) δ13C (†data from Exhaleair Cave and Nettlebed Cave from Hellstrom et al.[68]); and Dave's Cave flowstone (**h**) Mg/Ca (**i**) Sr/Ca, (**j**) δ44Ca; (**k**) δ18O and (**l**) δ13C.

revealed the presence of humic-like (peaks C and A)[54] (excitation: 240–270 nm (320–330 nm), emission: 420–480 nm) and protein-like (peak T) PARAFAC components (excitation: 250–280 nm, emission: 300–340 nm). At Dave's Cave, DOC fluorescence was characterised by a single humic-like component (i.e., no protein-like fluorescence observed) dominated by peak C (excitation: 330–350 nm, emission: 420–480 nm).

### ~14,000 years of DOC dynamics and environmental change
We present a reconstruction of DOC dynamics over the last ca. 14 kyr (Fig. 4), and compare our results to an alkenone-based sea-surface

temperature reconstruction from the Tasman Sea to assess the impact of temperature on DOC fluctuations (Fig. 4g)[38].

### 14,000–12,500 years: onset of flowstone accumulation
Aotearoa transitioned from the late-glacial into the Holocene between ca. ~18 and ~12 ka[85]. Owing to late glacial warming, flowstone accu-mulation restarted in many Te Waipounamu caves[70], including our study sites. From 14–12.5 kyr, the alkenone time-series indicates that sea-surface temperature rose sharply from 15.6 to 16.6 °C[38]. At the same time, humic-like DOC in Hodge Creek Cave increased by ~80%, from 4.5 to 8.1 mg C l⁻¹ (i.e., almost double the mean modern value).

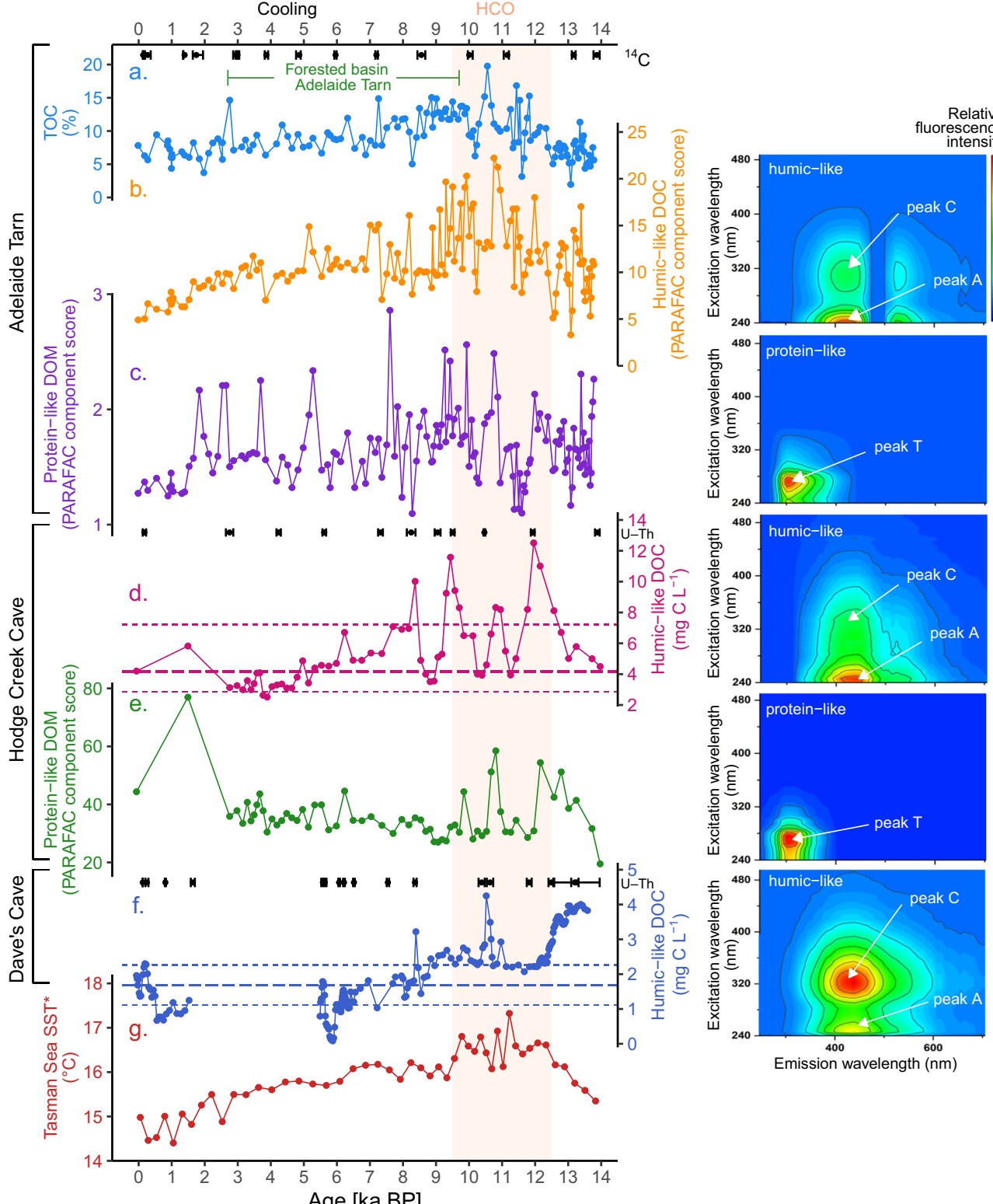

**Fig. 4 | ~14,000 years of DOC dynamics and environmental change.** Time-series data for (**a**) Adelaide Tarn Sediment TOC (inferred from FTIRS); **b** water-extractable humic-like DOC PARAFAC score (**c**) and protein-like PARAFAC score; **d** Hodge Creek Cave reconstructed dripwater-DOC concentration; **e** Hodge Creek Cave protein-like PARAFAC component score; **f** Dave's Cave reconstructed dripwater-DOC concentration; **g** *alkenone-derived Tasman Sea SST record (SO136-GC11)[38]. Dripwater humic-like DOC concentrations were reconstructed from the flowstone cores using humic-like DOC PARAFAC component scores and sample-specific $K_d$ values. Age results and 2σe (¹⁴C for Adelaide Tarn and U–Th for the flowstone cores) are shown in black. The mean and range of modern dripwater DOC concentrations are plotted (dashed lines). Where 3D EEM fluorescence was undertaken, corresponding excitation-emission matrices (EEMs) for each PARAFAC component are shown, with individual fluorophores labelled following Coble et al.[54].

The contemporaneous response to deglacial warming at Adelaide Tarn was less pronounced. While input of allochthonous inorganic material was high (as suggested by relatively high sediment magnetic susceptibility (Fig. 4a))[65], humic-like DOC and TOC oscillated strongly, likely due to active erosion of soil material from poorly-vegetated slopes[65]. Conversely, at Dave's Cave, DOC concentrations were at their highest for the entire flowstone record, with values > 200% higher than mean concentrations recorded during contemporary dripwater monitoring, whilst values from Mg/Ca, Sr/Ca and $\delta^{44}$Ca indicate higher PCP and relatively dry conditions. Elevated DOC at Dave's Cave could be associated with poorly developed soils, immediately following glacial retreat, as presumably soil development was slower compared to the other sites (owing to the relatively greater elevation and lower latitude of Dave's Cave).

### ~12,500–9500 years: elevated DOC export

The HCO was ~1–2.5 °C warmer than the immediate pre-industrial period[36,86,87] with 20–30% lower precipitation across most of Aotearoa[88]. Numerous palaeoenvironmental reconstructions indicate higher mean annual temperatures than present; however, lower treelines (including at Adelaide Tarn) indicate reduced seasonality (i.e., cooler summers and warmer winters)[36,89,90], possibly associated with lower summer insolation intensity and weaker westerlies in the Aotearoa sector of the Southern Ocean[91]. These conditions likely led to reduced orographic rainfall and ultimately drier conditions across northern Te Waipounamu. At Hodge Creek Cave, elevated Mg/Ca and Sr/Ca may indicate relatively drier conditions, an interpretation supported by more positive $\delta^{18}$O ratios (weaker amount effect) in flowstones from both Exhaleair and Nettlebed Caves[68]. Drier conditions are also consistent with a lake sediment reconstruction from southern Aotearoa, indicating periods of extended low lake levels (from 10–8 kyrs) attributed to diminished wind strength, higher air temperatures (as evidenced by increased biogenic silica), and reduced seasonality[92].

Over this interval, DOC concentrations oscillated at Hodge Creek Cave and Adelaide Tarn, but on average were high relative to the rest of the record, whilst at Dave's Cave, concentrations were high compared to the subsequent record but lower than the period from 14 to 12.5 kyrs (Fig. 4). In Hodge Creek Cave, DOC concentrations averaged 7.4 mg C l$^{-1}$, compared to the subsequent Holocene average (4.6 mg C l$^{-1}$), but oscillated substantially from 12.5 to 3.9 mg C l$^{-1}$. Results from Hodge Creek Cave are consistent with previous research on Mount Arthur speleothems, where at Nettlebed Cave, elevated $\delta^{13}$C (Fig. 3) and UV luminescence indicate increased soil productivity during the HCO[69]. Similarly, Adelaide Tarn recorded peak TOC and humic-like DOC values at this time, when its catchment was characterised by predominant grasses, shrubs and herbs[65], albeit with relatively high magnetic susceptibility values, likely owing to higher soil erosion, weak post-glaciation soils and an absence of trees within the catchment. In addition to vegetation changes, higher SOC production, solubility and biodegradation[60] (potentially driven by warmer temperatures) likely contributed to the elevated humic-like DOC concentrations in Hodge Creek Cave and Dave's Cave. Protein-like florescence was also elevated in Hodge Creek Cave and Adelaide Tarn over this interval, indicative of elevated microbial activity and DOC biodegradation (Fig. 4).

At Dave's Cave, DOC concentrations declined from 4.0 mg C l$^{-1}$ at 13 ka to 2.9 mg C l$^{-1}$ by 12.5 ka, possibly due to DOC dilution by glacial retreat and increased recharge prior to the onset of full Holocene interglacial conditions at ~11.5 ka[52]. Here, DOC concentrations continued to decline during overall wetter conditions (as indicated by Mg/Ca) between 12.5 and 12 ka (Fig. 4). From 12 ka, DOC values stabilised until a short interval of very high DOC at ~10.5 ka BP. The average DOC concentration during the HCO was ~45% higher than the rest of the Holocene record in this flowstone.

Despite higher mean annual temperatures, the plant macrofossil record from Adelaide Tarn suggests a lower-than-modern treeline through most of this period, wherein catchment vegetation was exclusively dominated by graminoids and bryophytes[36,65]. Based on palynological evidence from Adelaide Tarn, Jara et al. proposed that forest communities expanded upslope as a response to sustained warming from ~12.5 ka onwards, however, tree macrofossils first appear in the Adelaide Tarn core towards the end of the HCO at 9.7 ka[65], providing unequivocal evidence for the relatively late arrival time of trees in the catchment. The slow migration of trees into the catchment was attributed to high relief and rugged topography[65], or (despite higher mean annual temperatures) cooler summers and warmer winters (which can restrict treeline elevation, as reconstructed elsewhere on Te Waipounamu), a warmer ocean, and reduced westerly wind flow[65,86].

Given that both Adelaide Tarn and Hodge Creek Cave are positioned at lower elevation (1270 m and 940 m a.sl., respectively) than Dave's Cave ((1450 m a.s.l.) which is also located 4° further south)), we doubt the treeline was elevated above Dave's Cave during this period. Thus, elevated DOC concentrations during this period occurred under warmer conditions, with reduced seasonality[36] in unforested catchments at Adelaide Tarn (until 9.7 ka) and, very likely, at Dave's Cave. The latter also likely responding as a partial function of soil instability.

### The last ~9500 years: cooling and declining DOC export

The last ~9500 years was characterised by a continuous sea surface temperature decline in the Tasman Sea (Fig. 4)[38]. Following the HCO, the cooler and wetter mid-Holocene period (from 9.5–4 ka) began, during which sea-surface temperatures declined from 16.1 °C to 15.6 °C. This trend led to stronger south-westerly winds and wetter conditions between 9.5 and 7.5 ka[91] (Fig. 3).

Humic-like DOC concentrations declined at all three sites over this period, with declines of 41% and 43% at Hodge Creek Cave and Adelaide Tarn, respectively. The trend towards lower DOC concentrations in Adelaide Tarn followed the appearance of beech (*Fuscospora cliffortioides*) in the catchment, which displaced grasses and shrubs as the dominant vegetation, subsequently persisting for ~7000 years (~9.7–2.7 ka)[65]. Magnetic susceptibility declined markedly at ~7.8 ka (potentially owing to increased soil stability due to the presence of trees within the catchment), though values generally increased from 7 to 2 ka. Declining DOC concentrations also occurred at Dave's Cave from ~9 ka BP until a hiatus at ~5.5 ka, with calcite deposition resuming at ~1.5 ka. Although it is not possible to firmly establish the cause of the hiatus at Dave's Cave, it could be due to calcite undersaturation and lack of carbonate deposition. However, this conflicts with findings from several other palaeoenvironmental records, which provide strong evidence for cooler and more humid conditions[65] including elevated water tables in peatbogs (from 7 to 3.4 kyrs ago)[93] in the southern part of Te Waipounamu.

Through the last 4 kyrs, multiple lines of evidence indicate a sustained cooling in Aotearoa[38,65]. In the Tasman Sea, sea-surface temperature decreased from 15.6 °C to 15.0 °C[38], and the treeline again retreated from Adelaide Tarn's catchment some ~2.7 kyrs ago[65], which preceded a notable increase in magnetic susceptibility from 2.3 ka, which was sustained until 1 ka. During this interval, DOC reconstructions varied across sites, but consistently evidenced concentrations well below the long-term average (Fig. 4). From 4000 years to present, the Mg/Ca and Sr/Ca data indicates a general drying trend up to present at Hodge Creek Cave, an interpretation that is consistent with $\delta^{18}$O records from Exhaleair and Nettlebed caves[68] (Fig. 3).

### The influence of air temperature on DOC export

The reconstructed DOC fluxes over the Holocene reveal broadly coherent trends between cave sites (Hodge Creek Cave, Dave's Cave) and one lake (Adelaide Tarn) across four degrees of latitude.

Reconstructions of past DOC trends for both Hodge Creek Cave and Adelaide Tarn are remarkably consistent (Fig. 4). Given the proximity and similar altitudes of these sites, these similar DOC records provide evidence that DOC reconstructions presented herein reflect broad, landscape-level change. Nevertheless, the interactions and responses of multiple ecosystem processes and soil DOC export to climatic change are extremely complex and often non-linear[31]. Although we are unable to decipher whether temperature-mediated effects on organic matter production, decomposition rates or soil stability was the main driver of DOC export, the whole ecosystem response was for DOC export to increase during periods of elevated temperatures. At Hodge Creek Cave and Adelaide Tarn, humic-like DOC export peaked during the HCO (~12.5–9 ka BP), which had a mean annual temperature of 1.5–2.5 °C warmer than the late pre-industrial, with reduced seasonality[36,91], conditions that are analogous for future warmer-than-present climate[35,94]. Our findings are consistent with previously reported pollen-based reconstructions, which indicate the early Holocene was the most favourable time for forest growth and expansion, with conditions becoming gradually less favourable through the Holocene[68,95]. Although Adelaide Tarn's catchment was dominated by graminoids and bryophytes through most of the HCO, trees finally arrived in the catchment at 9.7 ka (with the lag-time attributed to slow migration rates or Adelaide Tarn's rugged, steep terrain[65]). Given Hodge Creek Cave's lower altitude compared to Adelaide Tarn (940 m a.s.l. and 1270 m a.s.l., respectively), it is likely that the treeline ecotone expanded above Hodge Creek Cave earlier than at Adelaide Tarn.

Our reconstructions for this period show DOC concentrations up to 2.5 times higher at Dave's Cave, and up to 3 times higher at Hodge Creek Cave than present-day. Furthermore, following the HCO, DOC concentrations declined at Adelaide Tarn and Hodge Creek Cave in tandem with sea-surface temperatures in the Tasman Sea[38]. From 10–7 kyrs, cooling drove a contraction of the conifer-broadleaf forest community in northwestern Te Waipounamu (the location of both Hodge Creek Cave and Adelaide Tarn)[65], although this trend lacked the intensity or continuity to drive the treeline from Adelaide Tarn's basin. The prolonged decline of DOC and TOC at Adelaide Tarn (despite the persistence of trees in the basin from ~9.7–2.7 ka) provides evidence that temperature-driven processes rather than the presence or absence of trees in the catchment was the primary control on soil DOC export. The positive relationship between temperature and aqueous DOC concentrations in the palaeoenvironmental records may be explained by combined temperature-driven organic carbon production[4,96,97] and degradation mechanisms[98]. Elevated temperature is an important mediator of net primary productivity[99], vegetation density[99], and microbial activity[100] including decomposition of primary substrates[100,101].

Although other factors such as hydrology and rainfall may impact DOC export, the clear correspondence of DOC concentrations with temperature at each site (apart from the immediate post-glacial period at Dave's Cave) provides firm evidence for temperature as the principal driver of soil DOC export. Temperature-driven DOC export in the certain absence of human interference (until ~1200 CE) is consistent with a study of contemporary (late twentieth and early twenty-first century) DOC export from undisturbed peatlands, where increased DOC export was attributed to warming (unlike disturbed peatlands, in which DOC export was controlled by recharge outflow)[29]. Further, several studies have demonstrated that warmer winters are associated with increased soil DOC productivity and export[102–104]. For example, an assessment of 1041 Swedish boreal lakes found that air temperature and DOC concentrations had a non-linear relationship, but that elevated DOC concentrations were strongly positively correlated with the number of positive-degree days[105]. Warmer winters reduce the depth and duration of snowpack, which provides thermal insulation to soils, thus exposing alpine soils to increased freeze-thaw frequency and severity. In one study, this was found to cause large (27%) increases in soil pore-water DOC concentrations[103]. DOC release under freeze-thaw conditions has been observed in field and laboratory conditions[103] and occurs due to the disruptive impacts of soil freezing on plant litter, plant roots, soil macro-aggregates and microorganisms.

## Uncertain influence of effective rainfall and hydrology

Leaching of soil DOC requires rainfall volumes that exceed field capacity[43]. However, the influence of rainfall amount on exported DOC loads and concentrations is complex, with several competing and complementary processes. For example, during drying events, higher DOC export can result from evapotranspirative concentration and increased microbial DOC-decomposition (compared to saturated conditions)[106,107], whilst soil drying and subsequent rewetting can drive increases in DOC release via priming of microbes, leading to sustained periods (i.e., years) of elevated DOC export[106–108].

Notably the relationship between rainfall and DOC is inconsistent between the Hodge Creek Cave and Dave's Cave archives. At Hodge Creek Cave, elevated Mg/Ca and Sr/Ca values provide some evidence for drier conditions during the HCO, while high DOC was evident at both Hodge Creek Cave and Adelaide Tarn during this period. However, at Dave's Cave, highly elevated DOC concentrations were recorded under drier conditions shortly after the onset of flowstone accumulation following glacial retreat from above the cave. Contrary to the trends recorded at Hodge Creek Cave, PCP declined alongside DOC at Dave's Cave, although DOC concentrations were still high compared to contemporary values.

Although the temporal resolution of our samples is too coarse to allow for reconstructions of individual hydrological events, the occurrence of periods of drought at Hodge Creek Cave and Adelaide Tarn during the HCO could be consistent with the elevated and oscillating DOC concentrations and higher PCP reconstructed over this period. There are several mechanisms by which periodic drying events (associated with low DOC export from soil) and subsequent rewetting of soil may lead to increased DOC export. Such an effect could be plausibly explained by the 'enzymatic latch' mechanism[109], whereby decomposition is suppressed by the presence of phenolic compounds. Under excessively wet conditions, oxygen supply for decomposition is reduced, slowing DOC degradation[110], but under dry conditions, a lower water table promotes oxic soil conditions, effectively eliminating phenolic compounds and their inhibitory effect on hydrolase enzymes[111,112]. Drier soils also favour lower DOC solubility, enabling organic matter storage, which can be released as DOC upon rewetting, somewhat counterintuitively increasing export of DOC[111,112].

Numerous studies have demonstrated that aquatic DOC concentrations can remain elevated for years after rewetting following individual drying events[109,111,112] because rewetting can stimulate the microbially-mediated breakdown of organic matter[108]. Thus, drying events followed by rewetting and the 'enzymatic latch' mechanism may explain sustained elevated DOC concentrations during dry periods at Hodge Creek Cave and Adelaide Tarn. However, DOC concentrations did not show any relationship with lower effective rainfall at Dave's Cave. Thus, while the impact of rainfall on DOC export is unclear, Holocene DOC concentrations were generally at their highest during the HCO at all three sites indicating a primary role for temperature-driven processes in controlling long-term DOC export. We suggest this temperature DOC dependence reflects temperature-mediated changes in inputs (i.e. primary production) and degradation rates of SOC.

## Limitations

The soil carbon cycle is extremely complex, encompassing interactions between climate, hydrology, plants, microbial activity, and minerals[31]. Numerous temperature-sensitive ecosystem functions

interact and respond to one another and may also display a non-linear response to chronic warming (e.g., vegetation productivity and carbon fixation, vegetation type, soil structure, microbial communities, and functioning)[31,113,114]. Through the past 14,000 years, treeline elevations and vegetation density and diversity certainly responded to temperature changes[65,88,95], as did soil erosion rates[65], and presumably microbial functions (as evidenced by fluctuations in protein-like fluorescence intensity at each study site).

Although we measured humic-like DOC present in dripwaters and speleothems, the rate and extent to which DOC is processed and filtered during transport from soil to cave is poorly constrained[43]. Changes in DOC properties may be explained by the soil continuum model[12], whereby organic matter (including SOC) is considered as 'a continuum of degrading compounds' ranging from intact plant material to highly oxidised carbon in carboxylic acids[12]. In the context of speleothems, DOC properties can be altered by microbial degradation during transport from soil to cave[43]. In addition, although DOC concentrations are reliably recorded in the crystal lattice during precipitation as shown in laboratory studies[42], the effects of dynamic environmental conditions within a cave setting (e.g., pH, redox state, ventilation, microbial activity) are uncertain. Although we assessed the fluorescence intensity of the protein-like fraction present in the flowstone cores, the extent to which microbial activity degraded DOC during transport is unclear. Notably, a protein-like fluorescence signal (indicating microbial activity) was observed at Hodge Creek Cave, yet at Dave's Cave, a cooler cave at greater elevation, no protein-like fluorescence was observed, with the fluorescence signal dominated by humic-like DOC.

Further research should focus on paired analysis of soil DOC (ideally through replicated soil profiles, as soil responses to climate change can vary laterally) and dripwater-DOC. Paired analysis of soil vs. cave DOC would provide insights on preferential removal or microbial transformations of different DOC fractions between soil and cave. For example, more hydrophobic fractions of DOC are likely to be preferentially removed (owing to adsorption to mineral surfaces) in the vadose zone, meaning that hydrophilic DOC may be more prominent in cave environments[115]. Similarly, in most karst environments, overlying soil is likely to be the main source of DOC to a cave[43]. However, there are some uncertainties in our assumption that humic-like DOC represents soil DOC. For example, soil-derived DOC could have previously been exported and stored in the vadose zone and then periodically mobilised into the cave (e.g., via hydrological or geochemical processes such as desorption from mineral surfaces), potentially leading to over-estimations of soil DOC export at those time intervals. Further, there are other potential sources of humic-like DOC (e.g., vegetation and vadose zone sediments, biofilms) which may have contributed to the DOC pool measured at the cave sites.

There are several uncertainties and limitations to our approach for reconstructing humic-like DOC concentrations in cave dripwaters. For example, for each cave, partition coefficient ($K_d$) values were calculated from DOC concentrations (measured via conventional DOC analysis as well as fluorescence and PARAFAC analysis to derive humic-like fluorescence intensity) in modern dripwaters against the average humic-like DOC concentration measured in the upper 3 mm (i.e., a total of three samples at 1 mm intervals measured using fluorescence) of flowstone samples. Our dripwater monitoring programme was undertaken across two summers, yet the upper 3 mm of flowstone accumulated for ~2720 years at Hodge Creek Cave and ~19 years at Dave's Cave. To refine our approach for reconstructing past dripwater DOC, more extensive monitoring of DOC in dripwaters and DOC incorporated into calcite accumulating over the same period would be required. However, this approach would be difficult at our study sites owing to their relative inaccessibility and their slow contemporary flowstone accumulation rates (particularly at Hodge Creek Cave).

## Conclusions

In conclusion, Aotearoa speleothems and lake sediments provide valuable resources for reconstructing the impact of climate on long-term terrestrial carbon dynamics. Observing long-term changes in soil DOC export is imperative, as whole ecosystem responses to climate-driven changes may take decades or centuries to manifest[31]. Authigenic carbonates have long been known to fluoresce owing to the presence of organic matter[116], and more recently have been shown to reliably record aqueous DOC concentrations in laboratory settings[28], demonstrating that speleothems reliably record changes in the abundance and molecular characteristics of DOC. Lakes are well-established archives of TOC and DOC[32,57,117], therefore we compared the Hodge Creek Cave and proximal Adelaide Tarn sediment archives, demonstrating consistencies that reflect regional, climate-driven changes. Aquatic DOC concentrations were substantially higher in the geologic past, most notably during the HCO, a warmer period with reduced seasonality where conditions were favourable for increased net primary productivity, forest expansion[68,95], and decomposition of primary substrates[100,101]. Following the HCO, temperatures declined[38] and DOC export showed a sustained declining trend through the mid-late Holocene.

In simple terms, soil C stocks reflect the balance between inputs and outputs (DOC export and decomposition), with both processes influenced by climate[3]. Our findings from fossil DOC archives are consistent with previous studies which identified that increasing temperature is an important driver of DOC export, leading to correspondingly higher concentrations in surface waters[28,29,118] and groundwater[14]. Given the global temperature increases already observed since the industrial revolution (+1.1 °C 1909 to 2019), aquatic DOC concentrations may already be increasing globally due to climatic forcing[28,98]. Future projections of mean annual temperature range from +0.7 °C to +3.7 °C by 2110 (relative to 1986–2005) in Aotearoa, with the greatest proportional warming at higher altitudes[119]. The relationship between DOC and drivers is often complex and non-linear[4,14] with cold and alpine environments being more sensitive to temperature change[3]. However, our analysis of paleoenvironmental archives suggests that future warming will ultimately drive increased DOC export from alpine and sub-alpine soils into aquatic systems, with implications for ecosystem functionality, water quality and the global carbon cycle[11,13,14].

## Methods
### Flowstone sample collection and analytical methods
Intact flowstone cores were collected in February 2015 from Dave's Cave and Hodge Creek Cave using a diamond-tipped coring bit powered by an 18 V electric drill. Prior to sampling, flowstones were inspected for the likelihood of good preservation, lateral laminations, and for active growth. Each sample was extracted from sites away from the cave entrance to reduce the likelihood of contamination. For cave preservation purposes, holes created by coring were plugged with pre-cut calcite discs. In preparation for analysis, flowstones were encased in resin and cut into two vertical slabs of equal thickness, exposing a stratigraphic face which was polished and used for sub-sampling and analysis.

This study employs 3D excitation-emission matrix (EEM) fluorescence spectroscopy[52–54], an established technique for DOC monitoring and characterisation based on the fact that fluorescence intensity is a linear function of DOC concentration within the concentration range observed in groundwater[14]. Through the sample preparation process, caution was observed to minimise the risk of sample contamination[120]. For example, prior to sub-sampling, samples and equipment were cleaned thoroughly with 95 % ethanol. Sub-sample powders were micro-milled at 1 mm resolution (Dave's Cave $N = 181$; Hodge Creek Cave $N = 67$) along the growth axis of each flowstone. Milling was undertaken using a tungsten carbide drill-bit. Powdered calcite (5 mg)

was dissolved in 4.5 ml of ultrapure 0.025 M HCl (18 MΩ water) in acid-washed (10 % HNO₃) polypropylene tubes and then centrifuged at 2608 g for 20 min. The resulting solutions had a pH of ~5.6, ensuring that all CaCO₃ was digested and a stable pH, typical of soil environments, was obtained.

Through each flowstone archive, filtered (0.45 μm) samples of dissolved flowstone calcite were measured for fluorescence using 3D EEMs (methods are described in detail in Pearson et al.[42]). EEM spectra combined with parallel factor analysis of components (PARAFAC) enables quantification of DOC and characterisation (and therefore source delineation) of the constituent molecules[121]. For example, this approach distinguishes humic-like, soil-derived material from autochthonous algae-derived DOC, which is characterised by a higher proportion of amino acids[122]. In soil leachates, the strong relationship between humic-like fluorescence and DOC concentrations is well-known[123], and has been used to assess DOC concentrations in alpine lake waters[124], speleothems[42], and lake sediments[56].

### U–Th geochronology of flowstones

Micro-milling (Sherline 500 vertical milling machine with a 0.5 mm tungsten carbide drill bit) of powder samples (weighing 100 mg) was undertaken on sectioned flowstone core samples perpendicular to the growth axis (Supplementary Figs. 3 and 5). The typical width of samples was 2 to 4 mm, with a thickness (i.e., z-axis) of 2 to 3 mm. The dimensions of each trench were recorded to allow estimation of depth uncertainty when calculating ages[125].

The ratios for $^{230}$Th/$^{234}$U age calculations were measured using a Nu Instruments multi-collector inductively coupled plasma mass spectrometer at the University of Melbourne, using the method described in Hellstrom[126]. Age-depth modelling was undertaken using the Constructing Proxy Records from Age Models (COPRA) algorithm[125] in MATLAB. COPRA was used to generate 2000 Monte-Carlo simulations of each age-model, whilst Piecewise cubic Hermite interpolation (PCHIP) was applied to interpolate the ages and produce median proxy values with 95% confidence intervals.

### PARAFAC models of fluorescent dissolved organic carbon

Samples (dissolved flowstone calcite and lake sediment water extractions) were placed in quartz cuvettes (4 ml volume, 1 cm width) and measured for fluorescence using methods outlined in Pearson et al.[42]. The fluorescence protocol used a 0.5 s integration time, a step-size of 3 nm, and a measurement range of 600–240 nm excitation and 800–245 nm emission. To correct for instrument specific biases, each matrix was corrected for inner-filter effects, scatter lines were Rayleigh masked, and spectra were then Raman normalised to the mean Raman intensity of Milli-Q water (18.0 MΩ) using the instrument's in-built software. 3D EEMs were processed using PARAFAC using the N-way toolbox[127], a multivariate modelling technique developed in MATLAB®. PARAFAC provides multi-way data analysis in which the underlying phenomena of the fluorescence can be distinguished and separated into statistically valid independent components, thus providing estimates of the relative contribution of each component to the total fluorescence signal. PARAFAC enables quantification of common fluorophores present in natural organic matter samples (i.e., humic-like and protein-like peaks) as statistical components[33]. Humic-like fluorophores include 'peak C', which represents allochthonous organic carbon in aquatic systems, whilst 'peak A' represents terrestrial humic substances[128]. Protein-like fluorescence is autochthonous in origin and characterised by peak T and was not utilised to reconstruct humic-like DOC export.

The quantitative fluorescence intensity (QFI) scores of the humic-like DOC components for each site were then used to reconstruct changes in total humic-like fluorescence over time. For the speleothem sites, dripwater humic-like DOC was reconstructed using a site-specific DOC partition coefficient based on dripwater and speleothem fluorescence measurements.

For calibration of flowstone humic-like DOC concentrations using natural DOC standards, water was pumped from 20 m depth in Kopuatai peat bog, central North Island, Aotearoa New Zealand, and filtered through 0.45 μm cellulose-acetate syringe filters (Microanalytix Pty Ltd, Australia), prior to TOC analysis. The peat water was diluted with deionised water to produce solutions containing between 5 and 15 mg C l⁻¹ (as non-purgeable DOC) and was calibrated against the QFI of humic-like PARAFAC components, producing a strong positive calibration ($R^2 = 0.99$). The peat water contained one, simple humic-like component (with no protein-like fluorescence detected)[28]. To calculate humic-like DOC concentrations in the speleothems and dripwaters, 3D EEM data from each site (speleothems and dripwaters) were analysed in PARAFAC alongside Kopuatai peat water calibration EEMs to create a bespoke calibration for each site (as humic-like fluorescence characteristics were slightly variable between cave sites). The calibrations were then used to calculate humic-like DOC concentrations in the individual cave dripwaters and speleothem calcite sub-samples. For the lake sediment core, total humic-like fluorescence was reconstructed by summing the QFI values of the two humic-like components.

### Monitoring of dripwater DOC and comparison to speleothems

Cave monitoring and sampling in Hodge Creek Cave and Dave's Cave was carried out during visits in February 2015, February 2016, May 2021, and March 2022. Drip rate measurements (Stalagmate® acoustic drop counter; Driptych, UK) in Hodge Creek Cave at five sites at and around the flowstone sampling location indicate generally slow (<1 drip/min) but highly variable discharge, occasionally ceasing entirely for several weeks. In Dave's Cave, all four monitored drip points were continuously active, with discharge varying between 0.5 and 1 drip/min across 1 year at the site of the flowstone core. Narrow temperature variations (measured with HOBO® TidbiT temperature loggers; Onset, USA) in the sampled chambers of both caves attest to limited connectivity to the external atmosphere, with annual mean temperatures of 6.3 ± 0.6 °C and 4.1 ± 0.7 °C in Hodge Creek Cave and Dave's Cave, respectively.

Dripwaters collected from both caves during each visit were filtered at 0.45 μm with cellulose acetate syringe filters (Microanalytix Pty Ltd, Australia), before analysis for trace elements (via solution-ICP-MS, acidified to 2% v/v with double-distilled HNO₃), DOC (as DOC using a persulfate oxidation method), and 3D EEM fluorescence.

To reconstruct allochthonous dripwater DOC from speleothem DOC, a partition coefficient (between aqueous and carbonate phases) is required. Given that partitioning is sensitive to the intrinsic properties of the cave setting and fluid composition, we derived empirical $K_d$ values for each speleothem, based on analysis of modern dripwater and the upper 3 mm of each speleothem core. A representative DOC concentration of the modern dripwater was determined using both EEMS and an independent TOC measurement ($R^2 = 0.99$; $n = 7$). The partition coefficient for each site was calculated using Eq. 1, with modern average dripwater DOC and Ca concentrations substituted into the denominator and measurements from the top 3 mm of speleothem in the numerator:

$$K_d = (DOC_s/Ca_s)/(DOC_{aq}/Ca_{aq}) \qquad (2)$$

where $K_d$ = partition coefficient, $_s$ = within speleothem, $_{aq}$ = within dripwater, and DOC = humic-like DOC. All units are in moles.

This exercise returned comparable, but higher $K_d$ values ($\log_{10} K_d = 0.78–1.14$) to those observed in lab-based precipitation experiments with peat-derived DOC ($\log_{10} K_d = 0.43–0.63$), consistent with lower growth rates in these cold, alpine caves, versus the faster growth typical of experiments performed under laboratory conditions ($\log K_d$ values converge on 0 with increasing growth rate)[50].

The composition of fluorescent DOC, as identified by PARAFAC models varied from site to site. In the Hodge Creek Cave flowstone, component 1 (excitation wavelength 250–280 nm; emission wavelength 300–340 nm) was identified as protein/tryptophan-like while component 2 (excitation wavelength 250–260 nm; emission wavelength 380–480 nm) was humic-like. In the Dave's Cave flowstone, DOC fluorescence was characterised by a single humic-like component with excitation wavelengths between 330–350 nm and emission wavelengths of 420–480 nm. Adelaide Tarn sediments reveal two humic-like components and one protein-like fluorophore. As this study was focused on DOC export from soils, humic-like DOC was then reconstructed in each palaeo-archive.

Theoretical co-evolutions of dripwater Mg/Ca and Sr/Ca due to PCP as shown in Fig. 2a were modelled using the spreadsheet made available by Tremaine and Froelich[84]. Calculations are based on the lowest measured dripwater Sr/Ca value for each cave (0.85 mmol mol$^{-1}$ for Hodge Creek Cave, 0.60 mmol mol$^{-1}$ for Dave's Cave) and use empirical calcite partition coefficients derived from modern calcite and dripwater trace element measurements from both locations. Partition coefficients of Mg ($D_{Mg}$) and Sr ($D_{Sr}$) were 0.04 and 0.19 for Hodge Creek Cave, and 0.11 and 0.51 for Dave's Cave, respectively. Small potential contributions of between 2% and 8% dolomite in the bedrock as parameterised in the model[81] produced PCP envelopes encompassing all but two dripwater samples.

## Trace element measurements in flowstone cores

Trace element measurements from HC15-2 and DC15-1 were collected using laser ablation inductively-coupled plasma mass spectrometry (LA-ICP-MS). Analyses of DC15-1 were undertaken at The University of Waikato, and analyses of HC15-2 were performed at the University of Melbourne. For DC15-1, analyses of the speleothem surface were conducted by pulsing the laser at 10 Hz with a 100 μm spot size and energy density of 5.0 J/cm² with a scan speed of 27.855 μm/s. Helium was used as the carrier gas to transport the aerosol to an Agilent 8900 Triple Quad inductively Coupled Plasma-Mass Spectrometer (ICP-MS). Prior to sampling, each transect was traversed and cleaned by a rapid 100 μm spot size laser ablation cleaning sweep. Background counts (gas background, measured with the laser off) were collected for 30 s between samples. National Institute of Standards and Technology (NIST) glass SRM (610, 612) were analysed after every sample line to account for any instrument drift. Background counts were subtracted from the raw data, and all data were standardised to NIST 612 glass. NIST 610 glass was utilised as a secondary standard. GeoReM database[129] was utilised for NIST glass values.

Trace element analyses of HC15-2 was undertaken via LA-ICP-MS at the School of Earth Sciences, University of Melbourne using a 193 nm ArF excimer laser-ablation system coupled to an Agilent 7700 quadrupole ICP-MS. Prior to analysis of trace elements, a pre-ablation track (spot size of 154 μm, pulse rate 15 Hz) removed material to a depth of 10 μm to clean the sample surface. Analysis was undertaken by pulsing the laser at 10 Hz with a scan speed of 100 μm/s ablation was undertaken under ultra-high-purity helium, with the ultra-high-purity argon used to carry the ablated aerosol into the mass spectrometer. NIST 612 glass reference was used as an external standard, analysed three times with the same spot size and a scan speed of 15 μm/s.

For both flowstone samples, trace element data were post-processed using Iolite software (v3.32 (Paton et al.[130])). Trace element/Ca mass ratios were calculated assuming stoichiometry (40% Ca in CaCO$_3$).

## Analysis of oxygen and carbon isotopes through Dave's Cave flowstone core

Flowstone sub-samples were analysed for $\delta^{18}O$ and $\delta^{13}C$ values at the Godwin Laboratory, University of Cambridge, using a Thermo Delta V isotope ratio mass spectrometer coupled to a GasBench II on-line gas preparation/introduction system. For each sample, ~100–150 μg of carbonate was sealed in a borosilicate glass Exetainer vial with a silicone rubber septum and loaded into the Thermo Gasbench autosampler that holds 40 samples. Each batch of samples included 10 reference carbonates of the in-house standard Carrara Z (calibrated to VPDB using the international standard NBS 19) and two control samples of Fletton Clay. Samples and standards were first flushed with helium and then acidified with 104% orthophosphoric acid for 1 h at 70 °C and analysed with the Thermo Delta V mass spectrometer in continuous flow mode. Precision of Carrara Z was ±0.06 ‰ (1σ) or better for $\delta^{18}O$ and $\delta^{13}C$.

## Analysis of calcium isotopes through Dave's Cave flowstone core

Calcium isotope analysis of Dave's Cave flowstone core was undertaken at the University of Copenhagen. Flowstone sub-samples were dissolved in 2 M HNO$_3$, dried down and redissolved in 2 M HNO$_3$. Calcium in the sample solutions was purified from the matrix via a 2-step ion chromatography protocol following established procedures[131] shortened given the simple matrix of the samples. In detail, sample aliquots were loaded in 1 ml of 2 M HNO$_3$ onto a Biorad-type column with 2 ml of TODGA resin. After eluting the matrix in an additional 4 ml of 2 M HNO$_3$, the Ca-containing cut was collected in 5 ml of 15 M HNO$_3$. Subsequently, Ca-containing solutions were dried down and redissolved in 1 ml of 2 M HNO$_3$ for ion chromatography purification from Sr, which is not separated in the previous step. Sample solutions were loaded onto 3 cm long column made from a 1 ml squeeze pipettes filled with 200 ml Eichrom Sr-spec resin. Calcium was collected with the sample load and an additional 2 ml of HNO$_3$. Samples were then dried down and redissolved in 0.5 M HNO$_3$ for isotope measurements. Calcium isotope compositions were determined using a Neptune Plus housed at the Centre for Star and Planet Formation (University of Copenhagen) following a modified procedure from that described in Schiller et al.[131]. In short, samples were measured by standard sample bracketing relative to the NIST SRM 915b calcium standard. Individual analyses, consisting of 50 ×8.4 s integrations, were preceded by blank analyses consisting of 10 ×8.4 s integrations and all samples were analysed 5 times. Typical signals were 15 V on $^{44}Ca$ with an uptake rate of 50 ml/min and concentrations of around 3 ppm in medium resolution mode. Final stable isotope compositions are reported as the mean and 2 standard error of the five replicate analyses in the delta notation relative to SRM 915b as follows:

$$\delta^{42/44}Ca = \left[ \frac{\left(\frac{42_{Ca}}{44_{Ca}}\right)_{sample}}{\left(\frac{42_{Ca}}{44_{Ca}}\right)_{SRM915b}} - 1 \right] \times 1000$$

## Lake sediment core analyses

At Adelaide Tarn, two overlapping sediment cores were collected using a piston corer as part of a previous study[65]. The Bayesian age-model of the core was based on 16 radiocarbon dates using the SHCal13 calibration curve[132].

For 3D EEM analysis, DOC was extracted from the lake sediments using a water extraction method commonly used in soil analysis[81]. Ten mg of freeze dried, homogenised sediment was added to 7 ml of distilled-deionised (18 MΩ) water in a polypropylene tube and shaken vigorously for 60 min, then centrifuged for 30 min at 2608 × g. Traditionally, assessments of water extractable organic carbon are limited to alkaline extractions from sediments[12]. However, Lehmann and Kleber[12] suggested that analysis should focus on water-soluble (and therefore bioavailable) material, since an alkaline treatment at pH 13 ionises compounds that would not dissolve in a natural pH range (pH 3.5 to pH 8.5). Adelaide Tarn measurements are therefore based on water-extractable dissolved organic carbon.

To buttress our findings from the PARAFAC analyses of 3D EEM data, we applied a partial-least squares regression model (based on a training-set based of 141 sediment samples from 11 Aotearoa lakes[56] ($R^2 = 0.88$) (Supplementary Fig. 8) to infer TOC concentrations via FTIRS of Adelaide Tarn's sediments. For TOC measurements of training-set sediments, samples first underwent acid pre-treatment to remove carbonates, followed by catalytic combustion (900 °C, $O_2$) and separation, before analysis in a thermal conductivity detector [Elementar Analyser]. The same samples were also analysed via FTIR spectroscopy using a Perkin-Elmer Spectrum 100 spectrometer following methods outlined in Pearson et al.[56].

### Geological sample collection statement
Flowstone cores were sampled under a Research and Collection authorisation (45832-GEO) from the Department of Conservation. Flowstones were sampled (rather than stalagmites) to minimise the aesthetic impact on each cave and flowstone core holes were then filled using pre-cut calcite discs. Ongoing calcite deposition at each sample location will eventually obscure the core holes.

### Data availability
Source data are provided with this paper.

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

## Acknowledgements

We would like to thank Steven Newcombe (University of Waikato) for assistance in cutting and polishing the speleothem slabs. We would also like to thank Travis Cross for fieldwork support. Thanks to Ignacio A. Jara (Universidad de Tarapacá) for providing data from Adelaide Tarn. Thanks to Steph Mangan (NIWA) for assistance producing Figs. 1, 3 and 4. Thanks to James Rolfe and John Nicolson (Godwin Laboratory, University of Cambridge) for undertaking $\delta^{18}O$ and $\delta^{13}C$ analysis. Thanks to Carsten Meyer-Jacob (Université du Québec en Abitibi-Témiscamingue) for PLSR modelling of FTIRS-TOC data. Thanks to Brittany Marie Ward (University of Waikato) for assistance in preparing geochemical proxy data. This study was made possible by Marsden Fund Grant UOW1403. This project was additionally supported by the New Zealand Ministry of Business, Innovation and Employment (MBIE) through the 'Global Change through Time' programme (Strategic Science Investment Fund, contract C05X1702); a Rutherford Discovery Fellowship award to A.H. (RDF-UOW1601); and by the New Zealand Government's Strategic Science Investment Fund (SSIF) made available to A.R.P. from the ESR project 'Groundwater in a warming world: Assessing resilience, threats, and implications'. M.S. acknowledges support from the Villum Fonden (00025333) and the Carlsberg Foundation (CF18_1105).

## Author contributions

A.R.P. completed sampling and analysis of the underlying data for this manuscript and designed the research project with A.H. A.H. secured funding, carried out cave sampling fieldwork, contributed to lab-analysis, helped interpret the findings, contributed to manuscript preparation and supervised A.R.P. B.R.S.F. gave support in interpreting the results, and supervised A.R.P. J.C.H. helped with fieldwork and undertook uranium–thorium (U–Th) disequilibrium dating of the speleothems (University of Melbourne). M.J.V. collected the Adelaide Tarn core and assisted in interpreting the results, and supervised A.R.P. S.F.M.B. provided support in sampling the flowstone cores and helped interpret the results of the study. R.N.D. ran the laser ablation inductively coupled plasma mass spectrometer (LA-ICP-MS) (University of Melbourne) on sample HC15-2 and helped interpret our findings. S.N.H. processed and interpreted Mg/Ca and Sr/Ca data from each cave and produced Fig. 2. C.T.W. processed and interpreted $\delta^{18}O$ and $\delta^{13}C$ data. M.S. ran $\delta^{44}Ca$ analysis (University of Copenhagen). Each author contributed to writing this manuscript.

## Competing interests

The authors declare no competing interests.
