## [Peer Review File · Nature Communications]

Warming drives dissolved organic carbon export from pristine alpine soilsREVIEWER COMMENTS

Reviewer #1 (Remarks to the Author):

Pearson and co-authors present a study on reconstructing DOC export from soils, using speleothems and a lake sediment core from Aotearoa New Zealand.

The results are clearly novel and address an important question, as soil DOC export and its response to global warming are poorly constrained and understudied. This is particularly due to a lack of possibilities to investigate the lability of DOC to warming in absence of anthropogenic interferences. Thus, the development of a method and archives that allow us to go further back in time and study soil DOC export during past periods of warming without human influence is important and timely.

While I think this study is well thought through and the methods seem valid and solidly carried out, I have some major concerns with the interpretation of the results, which often seem overconfident and not backed up enough by the data. At this point, I don't see the data in the study allowing the conclusions meriting publication in Nature Communications. However, I have some suggestions for improvement that may be addressed during a revision and may lead to clarification and more robust interpretations.

1) The DOC concentration proxy from speleothems and lake records has been developed and calibrated in previous studies, particularly Pearson et al., 2020. I commend the authors for their terrific work with the proxy development and validation. However, I would like to see some more critical assessment of the uncertainties that affect the DOC proxy. For example, while the humic-like fluorescence component can be separated from other components such as protein-like, and therefore the soil-derived signal can be isolated from e.g., microbial components, I assume there is no way to isolate a signature that could come from soil-derived OM previously exported at depths below the soil? If such a component of OM existed and was periodically exported as DOC, could it lead to an over-estimation of the soil DOC export at these times?

What about the general uncertainties with the interpretation of fluorescence signals and the calibration to DOC amounts? What about uncertainty in the meaning of the humic-like component (i.e., does it always have to relate to soil OM or are there other potential sources?).

I really would welcome a more nuanced discussion of this proxy and the reliability of the results (particularly in terms of quantitative reconstruction) in the paper, since as it stands it seems overconfident and potentially misleading.

2) The interpretation of speleothem Mg/Ca as a tracer for effective precipitation is not well backed up by data or further references. While Mg/Ca can be a tracer for effective infiltration, it can also relate to other processes, such as water-rock interaction (variability in the host rock composition). Is there monitoring data from the two caves that can help determining if this is the case? Data from the host rock? Replication with other speleothems? In any case, the authors need to provide more background information on how the interpretation of Mg/Ca was derived. It would also be nice to see a comparison with modelling data, as a check whether their hydroclimatic interpretation reflects modelled conditions

(check Scheff et al., 2017, Journal of Climate for a nice example).

Moreover, it is unclear to me whether the results shown are based off one single laser ablation track? This could also lead to strong skewing of the results, based on lateral variability of Mg/Ca in the stalagmite.

Without long-term monitoring and more detailed analysis of the flowstones (e.g., more than one LA sampling track) this is a rather tenuous interpretation. In that case, the claim that warming is the dominant driver of DOC export from these soils, and not hydroclimate, would not be backed up by the data. Similarly, the interpretation of possible seasonality signals in the data seems mostly based off of speculation.

A clearer description of how the data was generated and what gives the authors confidence that this proxy reflects effective precipitation at both cave sites is needed, including reflection on what potential sources of uncertainty could be, and how they would affect the results.

3) In general, the interpretation often lacks specificity and the data to back up some of the claims appears insufficient. Some of the claims seem cherry picked, and often are based on 1-2 data points given the resolution of the records. I tried pointing them out in the list of line-by-line comments below. Again, toning down the language in the manuscript, providing more details as to what could be potential complexities or issues in the dataset, would lead to a much more interesting and robust paper.

Comments line by line:

line 145: fragment sentence "Hodge Creek Cave (41° S, 172° E) is located ~32 ..."

Figure 2: The different trends and shapes of the records are difficult to see in this figure. I would suggest at least to plot the "climate" records (precip and T) separately from DOC.

It's also difficult to relate the z-score here to the absolute numbers described in the text. I would like to see a second version of the figure with absolute values.

line 188 and following: I find the separation of the three periods a bit confusing. Here, the peak described seems to already be in stage 2 at about 12ka? (Difficult to be sure because of the axis being in z-score and text in absolute numbers).

Either way, it seems to me that the Hodge creek cave record also mainly oscillates. There is a very high DOC value at 12 ka, but then values decrease again to levels comparable to the glacial and close to the mean of today. So the wording here is misleading.

line 194: Why is there only Mg/Ca in Dave's cave for this period? The speleothem at Hodge Creek cave appears to have been growing at that time?

line 224: is this short interval of very high DOC samples based on a single sample? Does the average of the HCO include this peak? What if this is an outlier, how would the values of the HCO without it compare to today?

line 232-234: Could the appearance of trees later in the record hint towards trees retaining more DOC, for example by reducing soil erosion? It would be nice if the authors could elaborate on whether, rather than only temperatures, the ecosystem change would affect DOC export.

line 369: Could the handling of the sample, embedding in resin and EtOH wash affect the fluorescence signal?

Reviewer #2 (Remarks to the Author):

Reviewer comments: Warming increases dissolved organic carbon export from pristine alpine soils

There is plenty of novelty in this manuscript, where the authors present records of organic carbon fluxes over the last ~14,000 years from duplicated speleothems and a lake sediment core. This is the first such speleothem-based reconstruction, and the evidence is of higher organic carbon fluxes at these alpine sites in the early Holocene, declining to the present day.

The authors propose a direct link with warming through comparison with sea surface temperature reconstructions for the region. However, there was also vegetation change over time as discussed in the manuscript. And also, soil formation and soil stability will change over time (e.g. Retallack, 2021), as the region recovered from the last glaciation. For this reviewer, these two alternate hypotheses (H: that the change in DOC export is due to changes in OC supply due to vegetation change; H: that the change in DOC export is due to soil formation and soil stability changes over time) need to be tested and refuted for the case to be made that the DOC export trends are due to temperature alone. Some evidence is presented for vegetation change, with evidence for the tree line changing over time and it seems that land cover and vegetation change would have occurred at the sites. Would additional trace element profiles using LA-ICP-MS help provide additional information on soil stability? I would add that in my opinion, the novelty of using speleothem and lake archives to reconstruct DOC over time is huge, irrespective of whether it is temperature, vegetation change, soil formation or something else.

A major novelty here is the use of speleothems to derive a proxy record of DOC. The approach, using fluorescence EEMs on dissolved powders, builds on Pearson et al (2020), and here uses a calibration against deep peat water extracts to quantify DOC in the speleothems in the past. Because of the novelty of the method, I would recommend that the calibration curve is provided in the Supplemental for the reader to assess. Also, the speleothem EEMs from the PARAFAC model are not shown, yet the end-members are. Again, it would be good to see this in the supplemental, so that the reader can see how alike the proxy EEMs are similar to the peat extracts end members.

The authors use a particular type of speleothem - flowstone – which is deposited at the base of flowing water. Flowstones are rarely used for paleoclimate research – stalagmites are favoured as their

stratigraphy is simpler and their hydrology relatively constrained (at least to water dripping onto their surface). The use of flowstone becomes relevant when the use of Mg/Ca ratios is introduced on Line 170. The authors state that this is a rainfall proxy, but this has to be demonstrated. As stated by the authors, it is a proxy for the time the water has previously had to precipitate calcite. For a flowstone, this could be along the flow path in the cave, and weakly or unrelated to rainfall. The authors provide duplicate flowstones, which show a lack of agreement in Mg/Ca over time in the early part of the records of the two samples. This suggests that it is not a direct rainfall proxy, and instead there is a karst hydrology control on the water evolution and flow paths for these two flowstones. This is important as the reconstructed DOC records are fluxes (e.g. mg/L) and it seems this is a combination of both source (supply) and flux (dilution). Again, the use of flowstones might be disadvantageous here, as they likely have higher water flux variability compared to stalagmites.

As far as I could tell, the authors do not consider or quantify the possible contribution of OC in the speleothems from the in-cave microbial community that will be present on the flowstone surface. This might be a small contribution relative to the soil, but should be addressed. If the PARAFAC components are presented, the reader would be able to assess whether there is a microbial component. This is particularly important as I believe Pearson et al (2020) previously suggested there is a microbial OM fluorescence signal in a NZ flowstone. It would be useful to understand more about any in-cave contribution to the speleothem record, how it changes over time and between samples, and whether it also co-relates with the other PARAFAC and environmental parameters.

Finally, would the authors consider adding multi-proxy data to support their new fluorescence proxy. Further LA-ICP-MS analyses of metal-transported trace elements, for example, would help understand the nature of the DOC being transported to the cave. Analysis of flowstone colour would be another proxy, I think one proposed previously by Pearson et al. (2020)? If colour calibrates with the fluorescence proxy, could it also enable a high-resolution record to be derived in a similar way that the authors use FTIR for the lake proxy record.

Retallack Gregory J. (2021) Soil, Soil Processes, and Paleosols. In: Alderton, David; Elias, Scott A. (eds.) Encyclopedia of Geology, 2nd edition. vol. 2, pp. 690-707. United Kingdom: Academic Press.
[x.doi.org/10.1016/B978-0-12-409548-9.12537-0](https://doi.org/10.1016/B978-0-12-409548-9.12537-0)

Reviewer #3 (Remarks to the Author):

Review comments for: NCOMMS-23-30654-T

Warming increases dissolved organic carbon export from pristine Alpine soils.

On the whole, I like this paper. It tackles an important research question (drivers of aquatic DOC dynamics). It also takes a long term perspective (Holocene duration) compared to most other studies on this topic which consider monitoring records which are only of decades duration. There is also a good overall finding that the speleothem DOC record can be broadly correlated with records of sea surface temperature, thus lending impact to the paper. This is also important given the longevity of the record and the lack of anthropogenic activity to provide alternative drivers to DOC release dynamics. This finding will be of interest to a broad research audience, thus lending the paper impact and significance. The methods used throughout are sound.

However, I have concerns about the way the paper is presented, the sweeping interpretations derived from the trace elements presented, the seemingly contradictory nature of some of the records, and in some cases the lack of attention paid to existing research in the field. Each of these issues are explained further below.

1. The paper is presented on the understanding not only is there a temperature control on aqueous DOC dynamics, but there is also a rainfall control that can be clearly deciphered. However, this quickly begins to unravel as soon as the reader accesses the results section. I would suggest a minor re-structuring to redress these expectations for the reader.

2. The issue above is compounded through the sweeping use of Mg/Ca as a rainfall proxy with little consideration of the many alternative explanations that can be attributed to changing Mg/Ca in the speleothem record. The assumption seems to be (although the explanation for the reader is extremely poor), that prior calcite precipitation will alter the ratio of Mg to Ca, such that drier conditions raise the concentration of Mg relative to Ca. However, Mg/Ca ratios are impacted by many other factors. In the epikarst, concentrations of magnesium can be driven by incongruent dissolution, variable bedrock contact time (including contact with variable proportions of host dolomite), changing hydrological pathway, in addition to prior calcite precipitation. Within the cave, partitioning across the drip water-calcite interface can alter speleothem Mg concentrations via temperature, PCO₂, crystal fabric, and growth rate. Thus, interpretation of the Mg/Ca is often speleothem specific (let alone cave specific) and requires the support from other proxies to decipher the true nature of the Mg signal. This should include a thorough assessment of speleothem calcite $\delta^{18}O$, Sr concentration dynamics, co-variation between Mg and Sr, and other indicators (eg. Ba to eliminate growth rate effects) to thoroughly address the proxy signal interpreted from Mg/Ca in these two speleothems. There is plenty of literature available which addresses these drivers and competing effects. Unfortunately, these assumptions undermine all of the discussion regarding the effects of effective rainfall. It is thus little wonder that the data are difficult to interpret and seemingly competing effects between Mg/Ca and DOC are seen.

3. The discussion surrounding some of the records is also a bit contradictory. There is confusion in the discussion of stage 2 regarding whether Dave's cave had higher, or lower, concentrations of DOC than the preceding stage. This is purely confusion in writing style and needs to be cleaned up. Further, as it is apparent the two speleothem records have two opposite trends between DOC and rainfall this does not

make for a convincing conclusion regarding rainfall as a driver of DOC dynamics. Whilst I appreciate rainfall drivers of DOC are complex, the reader is not informed about this until later in the discussion. This leads to confusion and lack of confidence in the way the data has been interpreted. I suspect the use of Mg as a rainfall indicator has also led to much of this poor interpretation. It is also poor that drying trends (which incidentally are opposite to that expected) are identified post 6ka in the speleothem from Dave's cave, but most of this time period is covered by a hiatus (Supplementary information, Figure 6). Despite this invoked drying trend, the subsequent sentences then proceed to identify the hiatus as due to undersaturation of the drip waters in a climatically wet period. Confusing as currently written. Throughout, many of the correlations identified between records require statistical tests to prove these.

4. I also felt as though there were many pieces of literature which were extremely relevant to the current manuscript, but which had not been paid attention. These include the many ground breaking studies by Baker et al. who pioneered the use of fluorescence EEM's to infer organic content of speleothem calcite; the more recent work by Blyth et al pioneering the extraction and molecular analysis of organic compounds from speleothem calcite, and the issues of contaminant molecules raised by Wynn and Brocks, 2014. (Rapid Communications in mass spectrometry). The work by Webb et al. 2014 (Journal of Quaternary Science) that addresses DOC in speleothem calcite from Australia and links this to the different proxy indicators of pluvial and arid phases; the wealth of literature surrounding interpretation of Mg and Sr in the speleothem record as indicators (or not) of prior calcite precipitation (Fairchild, numerous references; Sinclair et al., 2012. Chemical Geology) should also be addressed. There is also very little reference to the work of Worrall et al in the opening stages of the manuscript, paying reference to the work they have undertaken highlighting changing DOC fluxes from peatlands over the monitoring period.

Overall, I felt as though this was a manuscript that promised much and could have been so good! The lack of integrity with proxy interpretation spoiled the key findings. There is much data interpretation still to do before the story can be presented for publication in a journal such as Nature Communications. I hope these comments help a bit and I hope that the science comes to fruition as it will be an exciting science story when it has been cleaned up and presented in a more thorough fashion.

Please see a point-by-point response to reviewers' comments (blue) which directly follow each referee comment (black). Relevant text pasted from the manuscript is in *grey italics*.

Reviewer 1:

Reviewer 1, comment #1: The DOC concentration proxy from speleothems and lake records has been developed and calibrated in previous studies, particularly Pearson et al., 2020. I commend the authors for their terrific work with the proxy development and validation. However, I would like to see some more critical assessment of the uncertainties that affect the DOC proxy. For example, while the humic-like fluorescence component can be separated from other components such as protein-like, and therefore the soil-derived signal can be isolated from e.g., microbial components, I assume there is no way to isolate a signature that could come from soil-derived OM previously exported at depths below the soil? If such a component of OM existed and was periodically exported as DOC, could it lead to an over-estimation of the soil DOC export at these times?

Response: We wish to thank the reviewer for their comments on our previous work. We were concerned that there may be potential for over-estimates of DOC owing to the processes described by the reviewer. Unfortunately, we do not have a method for separating freshly exported soil DOC from material previously exported below the soil zone.

Although our previous research demonstrated that calcite can reliably record humic-like DOC concentrations and characteristics in a lab setting, we were uncertain as to whether flowstones could be used to reconstruct DOC export in an environmental setting (e.g., owing to complexities regarding OM transport such as previously exported soil-derived OM present at depths below the soil and periodically exported from the vadose zone to the cave, or microbial degradation of DOC within the cave itself). Thus, we decided to compare one of our flowstone archives (Hodge Creek Cave) against a proximal lake sediment archive (lake sediments are a more conventional archive of OC). The relative similarities between the lake sediment and flowstone records give us confidence that (despite different DOC transport pathways), our results are robust, although we acknowledge the potential for error associated with the processes outlined by the reviewer. In essence, we consider that on the decadal-to-centennial resolution represented by our datasets, stochastic changes in DOC transport and storage within the epikarst and vadose zone would have been integrated by the speleothems and therefore were not an important part of the signal we observe in these records. We agree with the reviewer that it is important to highlight these uncertainties, and have added a new subsection of the discussion titled '*Limitations*' which includes the following text to highlight the need for further research on the contribution of soil OM previously exported to depths below the soil profile:

“Further research should focus on paired analysis of soil DOC (ideally through soil profiles, as soil responses to climate change can be different through a profile) and dripwater-DOC. Paired analysis of soil vs cave DOC would provide insights on preferential removal or microbial transformations of different DOC fractions between soil and cave. For example, more hydrophobic fractions of DOC are likely to be preferentially removed (owing to adsorption to mineral surfaces) in the vadose zone, meaning that hydrophilic DOC may be more prominent in cave environments¹¹⁰. Similarly, in most karst environments, overlying soil is likely to be the main source of DOC to a cave⁴⁰. However, there are some uncertainties in our assumption that humic-like DOC represents soil DOC. For example, soil-derived DOC could have previously been exported and stored in the vadose zone and then periodically mobilised into the cave (e.g., via hydrological or geochemical processes such as desorption from mineral surfaces), potentially leading to over-estimations of soil DOC export at those time intervals. Further, there are other potential sources of humic-like DOC (e.g., vegetation and vadose zone sediments, biofilms) which may have contributed to the DOC pool measured at the cave sites”.

Reviewer 1, comment #2: What about the general uncertainties with the interpretation of fluorescence signals and the calibration to DOC amounts? What about uncertainty in the meaning of the humic-like component (i.e., does it always have to relate to soil OM or are there other potential sources?).

I really would welcome a more nuanced discussion of this proxy and the reliability of the results (particularly in terms of quantitative reconstruction) in the paper, since as it stands it seems overconfident and potentially misleading.

Response: We propose that most humic-like OM present in the flowstones derives from overlying soils (including vegetation and litter which contributes to the SOC pool), given the presence of organic-rich soils overlying both caves (see Supplementary for images and descriptions of the soil profile). Nevertheless, we acknowledge that this is an assumption, and that there are other potential sources (e.g., vadose zone biofilms, DOC previously released and stored below the soil profile). We have proposed some future research on paired analysis of soil vs cave DOC (see response to previous question) and have highlighted these uncertainties and limitations in the discussion section:

“Although we measured humic-like DOC present in dripwaters and speleothems, the rate and extent to which DOC is processed and filtered during transport from soil to cave is poorly constrained⁴⁰. Changes in DOC properties may be explained by the soil continuum model¹², whereby organic matter (including SOC) is considered as ‘a continuum of degrading compounds’ ranging from intact plant material to highly oxidised carbon in carboxylic acids¹². In the context of speleothem science, DOC properties can be altered by microbial degradation during transport from soil to cave⁴⁰. In addition, although DOC

concentrations are reliably recorded in the crystal lattice during precipitation as shown in laboratory studies³⁹, the effects of dynamic environmental conditions within a cave setting (e.g., pH, redox state, ventilation, microbial activity) are uncertain. Although we assessed the fluorescence intensity of the protein-like fraction present in the flowstone cores, the extent to which microbial activity degraded DOC during transport and prior to calcite incorporation is unclear. Notably, a protein-like fluorescence signal (indicating microbial activity) was observed at Hodge Creek Cave, yet at Dave's Cave, a cooler cave at greater elevation, no protein-like fluorescence was observed, with the fluorescence signal dominated by humic-like DOC".

Reviewer 1, comment #3: The interpretation of speleothem Mg/Ca as a tracer for effective precipitation is not well backed up by data or further references. While Mg/Ca can be a tracer for effective infiltration, it can also relate to other processes, such as water-rock interaction (variability in the host rock composition). Is there monitoring data from the two caves that can help determining if this is the case? Data from the host rock? Replication with other speleothems? In any case, the authors need to provide more background information on how the interpretation of Mg/Ca was derived. It would also be nice to see a comparison with modelling data, as a check whether their hydroclimatic interpretation reflects modelled conditions (check Scheff et al., 2017, Journal of Climate for a nice example).

Response: We thank the reviewer for this helpful comment. We agree that our previous submission lacked several important references, and we acknowledge that we did not provide enough detail on how the interpretation of Mg/Ca was derived. We added the following text and Figure 2 into our 'Results and Discussion':

"Inorganic and prior calcite precipitation proxies in dripwaters and flowstones

Given the relative dearth of paleo-hydrologic data from Aotearoa, we developed a multi-proxy dataset based on speleothem prior calcite precipitation (PCP) proxies^{58,59}, which are increasingly applied in speleothem paleoclimate studies to indicate hydrologic change, in addition to more conventional oxygen ($\delta^{18}O$) and carbon ($\delta^{13}C$) isotope ratios. PCP proxies respond to the precipitation of calcite along the speleothem flowpath and broadly reflect drier, better-ventilated conditions within the epikarst, karst aquifer, and cave environment. In some speleothems, Mg/Ca and Sr/Ca provide insight on effective infiltration (and therefore effective rainfall) through PCP and other controls, which increase Mg/Ca and Sr/Ca ratios during drier periods⁷⁰. However, results can be confounded by local hydrological controls on water evolution and flow paths⁷¹. Calcium isotope ratios ($\delta^{44/42}Ca$) have emerged as a proxy for local infiltration^{72,73}, as the lighter isotope (^{42}Ca) is preferentially precipitated during PCP^{72,74}. Factors controlling oxygen isotopes ($\delta^{18}O$) are numerous and complex⁷⁵, including

sensitivity to cave temperature and effective precipitation (amount, or moisture source)^{76,69}, however more positive values may indicate lower rainfall when the amount effect is active. $\delta^{13}\text{C}$ behaviour in cave systems is also highly complex⁷⁷, and can be influenced by soil respiration⁷⁸, and in-cave processes such as drip rate and degassing⁷⁹. $\delta^{13}\text{C}$ variability in high-altitude speleothems has also been attributed to changes in vegetation cover and/or soil thickness⁸⁰, including in Mount Arthur flowstones (Figure 3)⁶⁵. However, $\delta^{13}\text{C}$ can also move positively due to PCP, and covariation with Mg/Ca can suggest PCP influence. Additionally, covariance between $\delta^{13}\text{C}$ and $\delta^{18}\text{O}$ may represent increased kinetic fractionation, another indicator of relatively dry conditions in the epikarst.

PCP proxies respond to drier conditions because aridity increases the potential for gas exchange along dripwater flow paths. While PCP proxies have been interpreted within quantitative frameworks elsewhere⁵⁹, a fully quantitative treatment requires empirical functions between moisture balance and PCP, and further necessitates long-term monitoring datasets that are beyond the scope of this study. However, atomic ratios of magnesium (Mg) and strontium (Sr) to calcium (Ca) in modern dripwater samples, both Mg/Ca and Sr/Ca in Hodge Creek Cave and Dave's Cave largely cohere with expected evolutions due to PCP (Figure 2a)⁸¹. In the flowstone records, this imprint of PCP is supported by the slopes of Mg/Ca versus Sr/Ca signatures (in natural logarithm space; Figure 2b) of ca. 0.64 and 0.81 for Hodge Creek and Dave's Cave, respectively⁷¹. Although additional processes (e.g., incongruent calcite dissolution) likely contribute to Mg and Sr signatures (particularly in Dave's Cave), these data support the interpretation of karst hydrology as the dominant control.

Figure 2- a) Atomic ratios of Mg and Sr to Ca in dripwater samples from Hodge Creek Cave (blue dots) and Dave's Cave (red dots). Solid coloured lines reflect respective linear fits. Dotted coloured lines indicate theoretical relationships between Mg/Ca and Sr/Ca as a function of PCP for each cave, calculated for a limestone bedrock with small contributions of Mg from dolomite⁸¹; see Methods for details. **b)** Flowstone Mg/Ca and Sr/Ca signatures (expressed as natural logarithms), including

all data shown in the timeseries in Figure 3. Solid lines reflect respective linear best fits. For both speleothems, the slopes of this relationship cohere with a dominant hydrological control due to PCP.

To buttress our interpretations from trace element records from Hodge Creek Cave, we include published oxygen ($\delta^{18}\text{O}$) and carbon ($\delta^{13}\text{C}$) isotope records from Exhaleair Cave and Nettlebed Cave^{65,66}, which (like Hodge Creek Cave) are positioned on Mount Arthur, albeit at lower elevations (685 m and 390 m a.s.l., respectively). From the Dave's Cave flowstone, we present Mg/Ca and Sr/Ca alongside $\delta^{44}\text{Ca}$, $\delta^{18}\text{O}$, and $\delta^{13}\text{C}$ (Figure 3). Allowing for these caveats, the hydrological proxy data at large show coherent trends and allow qualitative interpretations with which to test the controls on DOC variations presented in Figure 4. Evolution of hydrological changes are discussed in relation to DOC dynamics in the remainder of the text."

Reviewer 1, comment #4: Moreover, it is unclear to me whether the results shown are based off one single laser ablation track? This could also lead to strong skewing of the results, based on lateral variability of Mg/Ca in the stalagmite. Without long-term monitoring and more detailed analysis of the flowstones (e.g., more than one LA sampling track) this is a rather tenuous interpretation. In that case, the claim that warming is the dominant driver of DOC export from these soils, and not hydroclimate, would not be backed up by the data. Similarly, the interpretation of possible seasonality signals in the data seems mostly based off speculation. A clearer description of how the data was generated and what gives the authors confidence that this proxy reflects effective precipitation at both cave sites is needed, including reflection on what potential sources of uncertainty could be, and how they would affect the results.

Response: The results shown are based off one laser ablation track for each sample. However, to support our interpretation of Mg/Ca, we have now included Sr/Ca records from flowstones at both Hodge Creek and Dave's Cave. To buttress our interpretations from our trace element records at Hodge Creek Cave, we also show previously published stable oxygen and carbon isotope records from Exhaleair Cave and Nettlebed Cave^{1,2}, which (like Hodge Creek Cave), are positioned on Mount Arthur, albeit at lower elevations (685 m and 390 m a.s.l., respectively). From Dave's Cave, we show reconstructions of Mg/Ca and Sr/Ca alongside stable records of oxygen, carbon, and calcium isotopes, the latter providing the most direct measure of PCP among these proxies.

Unfortunately, there may have been a misunderstanding regarding our description of seasonality. The time-resolution for our speleothem and lake sediment records is too coarse for any analysis of seasonal (or even annual) signals, however we cite previously published research to suggest that changes in seasonality (e.g., higher mean annual temperature with cooler summers and warmer winters) could have impacted long-term soil DOC export. For example, McGlone et al., 2011

proposed that the HCO may have had 'reduced seasonality' (i.e., cooler summers and warmer winters)^{3,4} (e.g., due to increased net primary productivity, forest expansion and treeline elevation owing to longer growth seasons). Thus, we suggest that that reduced seasonality^{3,5,6} through the HCO may have contributed to ecosystem change (production and/or degradation of SOC) which led to increased DOC increased export, most notably during the HCO. We have pasted some relevant text from our results and discussion section:

"The Holocene Climatic Optimum (HCO, ~12.5–9 kyrs) was ~1–2.5°C warmer than the immediate pre-industrial period^{31,83,84} with at least 30% lower precipitation across most of Aotearoa^{53,56}. Numerous palaeoenvironmental reconstructions indicate higher mean annual temperatures than present; however, lower treelines (including at Adelaide Tarn) indicate reduced seasonality (i.e., cooler summers and warmer winters)^{85,86}, possibly associated with lower summer insolation intensity⁸⁷ and weaker westerlies in the Aotearoa sector of the Southern Ocean⁵⁴. These conditions likely led to reduced orographic rainfall and ultimately drier conditions across northern Te Waipounamu/South Island. At Hodge Creek Cave, elevated Mg/Ca, Sr/Ca and may indicate relatively drier conditions, an interpretation supported by more positive $\delta^{18}\text{O}$ ratios (weaker amount effect) in flowstones from both Exhaleair and Nettlebed Caves⁶⁵. Drier conditions are also consistent with a lake sediment reconstruction from southern New Zealand, whereby periods of extended low lake levels (from 10–8 kyrs) were attributed to diminished wind strength, higher air temperatures (as evidenced by increased biogenic silica), and reduced seasonality⁸⁸".

And:

"Despite higher mean annual temperatures, the plant macrofossil record from Adelaide Tarn suggests a lower-than-modern treeline through most of this period, wherein catchment vegetation was exclusively dominated by graminoids and bryophytes^{31,62}. Based on palynological evidence from Adelaide Tarn, Jara et al. (2015) proposed that forest communities expanded upslope as a response to sustained warming from ~12.5 ka onwards, however, tree macrofossils first appear in the Adelaide Tarn core towards the end of the HCO at 9.7 ka⁶², providing unequivocal evidence for the relatively late arrival time of trees in the catchment. The slow migration of trees into the catchment was attributed to high relief and rugged topography⁶², or (despite warmer mean annual temperatures) cooler summers and warmer winters (which can restrict treeline elevation, as reconstructed elsewhere on Te Waipounamu/South Island), a warmer ocean, and reduced westerly wind flow^{62,83}".

Reviewer 1, comment #5: In general, the interpretation often lacks specificity and the data to back

up some of the claims appears insufficient. Some of the claims seem cherry picked, and often are based on 1-2 data points given the resolution of the records. I tried pointing them out in the list of line-by-line comments below. Again, toning down the language in the manuscript, providing more details as to what could be potential complexities or issues in the dataset, would lead to a much more interesting and robust paper.

Response: We thank the reviewer for this comment and have edited and added text to our manuscript to reflect the nuances, complexities, and limitations of our findings, and have toned down our language.

However, we respectfully contest the assertion that some of the claims are based on one or two data points (e.g., relatively elevated DOC concentrations during the HCO are evident in both Hodge Creek Cave, Adelaide Tarn, and Dave's Cave). We agree that our figures did not clearly show our data points, and have amended Figure 4 so that individual data points are shown (rather than lines as shown previously) (see Figure 4 in manuscript).

Reviewer 1, comment #6: Line 145: fragment sentence "Hodge Creek Cave (41° S, 172° E) is located ~32 ..."

Response: Thank you- this is now changed.

Reviewer 1, comment #7: Figure 2: The different trends and shapes of the records are difficult to see in this figure. I would suggest at least to plot the "climate" records (precip and T) separately from DOC.

It's also difficult to relate the z-score here to the absolute numbers described in the text. I would like to see a second version of the figure with absolute values.

Response: We have now changed the figure to be clearer. To avoid repetition, we chose to amend our figures to show absolute values instead of z-scores for both Figures 3 and 4. For visual clarity, we have also plotted data for each proxy separately.

Reviewer 1, comment #8: Line 188 and following: I find the separation of the three periods a bit confusing. Here, the peak described seems to already be in stage 2 at about 12ka? (Difficult to be sure because of the axis being in z-score and text in absolute numbers). Either way, it seems to me that the Hodge creek cave record also mainly oscillates. There is a very high DOC value at 12 ka, but then values decrease again to levels comparable to the glacial and close to the mean of today. So the wording here is misleading.

Response: DOC concentrations oscillated through the HCO period at Hodge Creek Cave, and the reasons for such oscillations are unclear. Nevertheless, mean DOC concentrations were much higher than values recorded through the rest of the record, and are similar (in terms of oscillation but also relatively higher values) to those recorded in Adelaide Tarn's Lake sediment core.

We highlight this in the results and discussion section: *“Over this interval, DOC concentrations oscillated at Hodge Creek and Adelaide Tarn, but on average were high relative to the rest of the record, whilst at Dave’s Cave, concentrations were high compared to the subsequent record but lower than the period from 14 to 12.5 kyrs (Figure 4). In Hodge Creek Cave, DOC concentrations averaged 7.2 mg C L⁻¹, 63% higher than the subsequent Holocene average (4.4 mg C L⁻¹) but oscillated substantially from 12.5 to 3.9 mg C L⁻¹. Results from Hodge Creek Cave are consistent with previous research on Mount Arthur speleothems, where at Nettlebed Cave, elevated $\delta^{13}\text{C}$ (Figure 3) and UV luminescence indicate increased soil productivity during the HCO. Similarly, Adelaide Tarn recorded peak TOC and humic-like DOC values at this time, when its catchment was characterised by predominant grasses, shrubs and herbs⁶², albeit with relatively high magnetic susceptibility values, likely owing to higher soil erosion, weak post-glaciation soils and an absence of trees within the catchment. In addition to vegetation changes, higher soil organic carbon production, solubility and biodegradation⁶⁰ (potentially driven by warmer temperatures) likely contributed to the elevated humic-like DOC concentrations in Hodge Creek Cave and Dave’s Cave. Protein-like fluorescence was also elevated in Hodge Creek Cave and Adelaide Tarn over this interval, indicative of elevated microbial activity and DOC biodegradation (Figure 4)”.*

Reviewer 1, comment #9: line 194: Why is there only Mg/Ca in Dave’s cave for this period? The speleothem at Hodge Creek cave appears to have been growing at that time?

Response: Unfortunately this period of growth was not captured during our laser ablation analytical session.

Reviewer 1, comment #10: Line 224: is this short interval of very high DOC samples based on a single sample? Does the average of the HCO include this peak? What if this is an outlier, how would the values of the HCO without it compare to today?

Response: Our description of relatively high-DOC during the HCO is based on several samples. To improve clarity, for the DOC data, we have now changed the figures to include point markers (previously, these data were plotted as a line. As discussed in the manuscript, DOC concentrations (and FTIRS-TOC at Adelaide Tarn) oscillated through the HCO, but for each record, on average DOC

concentrations were higher than concentrations recorded outside (i.e., before and after) of the HCO except for the early record at Dave's Cave. Please see text copied in response to question 8.

Reviewer 1, comment #11: Line 232-234: Could the appearance of trees later in the record hint towards trees retaining more DOC, for example by reducing soil erosion? It would be nice if the authors could elaborate on whether, rather than only temperatures, the ecosystem change would affect DOC export.

Response: There is potential that trees reduced soil erosion, which could have affected DOC export. We are unaware of any proxies for soil erosion in speleothems but have included previously published (Jara et al., 2015) magnetic susceptibility data (as a proxy for erosion) from the same Adelaide Tarn sediment core (Figure 4).

We mention the magnetic susceptibility data in our discussion section: *"While input of allochthonous inorganic material was high (as suggested by relatively high sediment magnetic susceptibility (Figure 4 a))⁶², humic-like DOC and TOC oscillated strongly, likely due to active erosion of soil material from poorly vegetated slopes⁶². Conversely, at Dave's Cave, DOC concentrations were at their highest for the entire flowstone record, with values >200% higher than mean concentrations recorded during contemporary dripwater monitoring), whilst values from Mg/Ca, Sr/Ca and $\delta^{44}\text{Ca}$ indicate higher PCP and relatively dry conditions".*

And: *"Results from Hodge Creek Cave are consistent with previous research on Mount Arthur speleothems, where at Nettlebed Cave, elevated $\delta^{13}\text{C}$ (Figure 3) and UV luminescence indicate increased soil productivity. Similarly, Adelaide Tarn recorded peak TOC and humic-like DOC values at this time, when its catchment was characterised by predominant grasses, shrubs and herbs⁶², albeit with relatively high magnetic susceptibility values, likely owing to higher soil erosion, weak post-glaciation soils and an absence of trees within the catchment. In addition to vegetation changes, higher soil organic carbon production, solubility and biodegradation⁶⁰ (potentially driven by warmer temperatures) likely contributed to the elevated humic-like DOC concentrations in Hodge Creek Cave and Dave's Cave".*

We agree with the reviewer that ecosystem change (which is influenced by climate) would certainly have influenced DOC export. We make clear in the Introduction section that controls on DOC production and export are complex (though some are temperature dependent, particularly in the absence of human impacts) and influenced by numerous factors.

Soils represent the largest terrestrial pool of carbon (C) and store more carbon than terrestrial vegetation and the atmosphere combined^{1,2}. Soil organic carbon (SOC) stocks reflect a balance between numerous processes influenced by climate³, including net primary production⁴, microbial priming and decomposition rates⁵, and SOC solubility⁶⁻⁸. The release of dissolved organic carbon (DOC) from soils represents an important but uncertain flux within the global carbon cycle that remains poorly constrained, or not represented at all, in global carbon budgets⁹⁻¹¹.

Most DOC in freshwaters derives from plant tissue (following biological, physical and chemical processing in soil¹²). DOC export is thus directly linked to carbon storage in catchment soils, constituting the least constrained aspect of this mass balance (Eqn. 1)⁶.

$$SOC = C_{PP} - C_R - C_{DOC} \quad (1)$$

Where C_{PP} is the carbon fraction fixed by primary production, C_R is the respired carbon fraction (as CO_2), and C_{DOC} is the carbon fraction lost via DOC solubilisation and export.

Soluble carbon fluxes from soil are expected to form a 'climate feedback' as the world continues to warm⁵. Yet, the response of DOC export to rising global temperatures and changes in hydroclimate also has implications for freshwater and marine ecosystems. DOC increases in aquatic systems have significant impacts, including eutrophication⁷, reduced surface water clarity⁹, and enhanced contaminant transport⁸, as well as influencing groundwater pH¹³ and fuelling redox transformations in the subsurface¹⁴. Generally, aquatic DOC concentrations are positively related to SOC stocks in developed soils and vegetation cover¹⁵, but can also be influenced by the presence of peat, which can export up to 10 times more DOC than forest soils¹⁶.

Also in the introduction section, we note: "although individual processes may respond rapidly to climatic change, the whole ecosystem response to climate-driven changes may take decades or even centuries to manifest²⁷."

Reviewer 1, comment #12: Line 369: *Could the handling of the sample, embedding in resin and EtOH wash affect the fluorescence signal?*

Response: We are certain that embedding the sample in resin and EtOH wash did not affect the fluorescence signal. Resin was used to encapsulate each flowstone core, with the aim of ensuring that the samples remained intact when split in half with a saw to enable the face of the speleothem to be exposed. EtOH was used to clean the sample prior to coring of the sample. Once sub-sample milling commenced, the upper 0.2 mm of each sub-sample was disposed of, to ensure that there was no impact from any potential contamination on the surface of the exposed speleothem slab.

Reviewer 2:

Reviewer 2, comment #1: The authors propose a direct link with warming through comparison with sea surface temperature reconstructions for the region. However, there was also vegetation change over time as discussed in the manuscript. And also, soil formation and soil stability will change over time (e.g. Retallack, 2021), as the region recovered from the last glaciation. For this reviewer, these two alternate hypotheses (H: that the change in DOC export is due to changes in OC supply due to vegetation change; H: that the change in DOC export is due to soil formation and soil stability changes over time) need to be tested and refuted for the case to be made that the DOC export trends are due to temperature alone. Some evidence is presented for vegetation change, with evidence for the tree line changing over time and it seems that land cover and vegetation change would have occurred at the sites. Would additional trace element profiles using LA-ICP-MS help provide additional information on soil stability? I would add that in my opinion, the novelty of using speleothem and lake archives to reconstruct DOC over time is huge, irrespective of whether it is temperature, vegetation change, soil formation or something else.

Response: We thank the reviewer for their comment regarding the novelty of our manuscript.

We agree with the reviewer that DOC export is influenced by vegetation changes and potentially treeline shifts, which influences carbon inputs to soils, as well as degradation and solubilisation rates (which influence DOC export to aquatic systems)⁷. Similarly, soil erosion and pedogenesis post-glaciation may have led to increased DOC export (particularly at Dave's Cave, where DOC concentrations are relatively high following the onset of flowstone accumulation after glacial retreat). However, we propose that temperature was an important driver of these co-occurring and complex environmental changes, which led to changes in DOC export.

Text on higher DOC owing to slow pedogenesis at Dave's Cave:

“Elevated DOC at Dave’s Cave could be associated with poorly developed soils, immediately following glacial retreat, as presumably soil development was slower compared to the other sites (owing to the relatively greater elevation and lower latitude of Dave’s Cave)”

As far as we are aware, there are no published papers on the use of speleothem trace elements for reconstructing soil stability. Developing a new proxy for soil stability was beyond the scope of this research project, and producing a defensible proxy for soil stability is likely to be unachievable or tenuous with our dataset. However, we have now included a lake sediment-based proxy (magnetic susceptibility) from Adelaide Tarn for soil stability/erosion (Figure 3 of the manuscript). We refer to the magnetic susceptibility data (both in terms of the impacts of shifting treelines on soil stability, and the potential impact of soil stability on DOC export) throughout the results and discussion section.

Reviewer 2, comment #2 A major novelty here is the use of speleothems to derive a proxy record of DOC. The approach, using fluorescence EEMs on dissolved powders, builds on Pearson et al (2020), and here uses a calibration against deep peat water extracts to quantify DOC in the speleothems in the past. Because of the novelty of the method, I would recommend that the calibration curve is provided in the Supplemental for the reader to assess. Also, the speleothem EEMs from the PARAFAC model are not shown, yet the endmembers are. Again, it would be good to see this in the supplemental, so that the reader can see how alike the proxy EEMs are similar to the peat extracts end members.

Response: We have also run PARAFAC models for each site independently to show the individual EEMs for each component from each site (including Kopuatai bog, which exhibits one humic-like component). As noted in the manuscript, there are some differences in the fluorophores between sites, as would be expected given the differing soil types and vegetation at each site. We now show PARAFAC component fluorophores for each DOC reconstruction (Figure 4).

Figure 4- Time-series data for **(a)** Adelaide Tarn Sediment TOC (inferred from FTIRS); **(b)** water-extractable humic-like DOC PARAFAC score **(c)** and protein-like PARAFAC score; **(d)** Hodge Creek Cave Reconstructed dripwater-DOC concentration; **(e)** Hodge Creek Cave protein-like PARAFAC component score; **(f)** Dave's Cave reconstructed dripwater-DOC concentration; **(g)** alkenone-derived Tasman Sea SST record (SO136-GC11)³³. The mean and range of modern dripwater DOC concentrations

are plotted for each record of dripwater humic-like DOC concentrations reconstructed from the flowstone cores using humic-like DOC PARAFAC component scores and sample-specific K_d values. Age results and 2σ (^{14}C for Adelaide Tarn and U–Th decay for the flowstone cores) are shown in black. Where 3D EEM fluorescence was undertaken, corresponding excitation-emission matrices (EEMs) for each PARAFAC component are shown, with individual fluorophores labelled following Coble et al., (1996)⁵¹.

In the Supplementary Information, we have also included the calibration between the fluorescence intensity of humic-like DOC (as measured following wet-oxidation TOC analysis). We also present the fluorophores for each reconstruction alongside the peat water extract (from Kopuatai bog) (Supplementary Figure 9 and Supplementary Table 4).

Supplementary Table 4- PARAFAC components and peak descriptions for each site (See Supplementary for protein-like components). For Adelaide Tarn, the QFI of each humic-like PARAFAC Component were summed to produce a value for humic-like fluorescence intensity, which were used to reconstruct DOC concentrations in the dripwaters and lake sediments.

Sites	PARAFAC Component #	3D Excitation-emission matrix (EEM)
Kopuatai peat dome	1	

Supplementary Figure 10- Fluorescence intensity of humic-like PARAFAC component against DOC concentration in water from Kopuatai bog.

Reviewer 2, comment #3: The authors use a particular type of speleothem - flowstone – which is deposited at the base of flowing water. Flowstones are rarely used for paleoclimate research – stalagmites are favoured as their stratigraphy is simpler and their hydrology relatively constrained (at least to water dripping onto their surface). The use of flowstone becomes relevant when the use of Mg/Ca ratios is introduced on Line 170. The authors state that this is a rainfall proxy, but this has to be demonstrated. As stated by the authors, it is a proxy for the time the water has previously had to precipitate calcite. For a flowstone, this could be along the flow path in the cave, and weakly or unrelated to rainfall. The authors provide duplicate flowstones, which show a lack of agreement in Mg/Ca over time in the early part of the records of the two samples. This suggests that it is not a direct rainfall proxy, and instead there is a karst hydrology control on the water evolution and flow paths for these two flowstones. This is important as the reconstructed DOC records are fluxes (e.g. mg/L) and it seems this is a combination of both source (supply) and flux (dilution). Again, the use of flowstones might be disadvantageous here, as they likely have higher water flux variability compared to stalagmites.

Response: We agree that we did not adequately demonstrate that Mg/Ca ratios could be used for precipitation reconstruction and acknowledge that there are several complexities and caveats associated with their use as rainfall proxies.

We agree that DOC supply and dilution may influence our DOC reconstructions in the speleothems. As such, at the start of this research project, we were uncertain as to whether speleothems could be used to reconstruct soil DOC export (calcite can reliably incorporate DOC from growth solutions, but the impact of transport processes from soil to cave are uncertain). Thus, we chose to compare one of our flowstone DOC records (Hodge Creek Cave) against a proximal lake sediment archive (Adelaide Tarn). We are confident that the similarity between DOC trends (despite different processes influencing transport incorporation into their respective archives) between the flowstone and lake sediment provides strong evidence that relative changes in DOC concentrations were reliably recorded. Although we acknowledge issues regarding use of flowstones, we selected flowstones because of the relatively minimal impact on the cave environment and aesthetics (compared to stalagmite sampling) and because they are fed by numerous feeding points, and therefore DOC concentrations may be more representative of DOC exported from a wider area (compared to stalagmites which are fed by one drip point).

Reviewer 2, comment #4: As far as I could tell, the authors do not consider or quantify the possible contribution of OC in the speleothems from the in-cave microbial community that will be present on the flowstone surface. This might be a small contribution relative to the soil, but should be addressed.

If the PARAFAC components are presented, the reader would be able to assess whether there is a microbial component. This is particularly important as I believe Pearson et al (2020) previously suggested there is a microbial OM fluorescence signal in a NZ flowstone. It would be useful to understand more about any in-cave contribution to the speleothem record, how it changes over time and between samples, and whether it also co-relates with the other PARAFAC and environmental parameters.

Response: We have now presented the PARAFAC EEMs and have plotted the fluorescence intensity of the protein-like (i.e., microbial-origin) EEMs in Figure 4. Protein-like fluorescence was recorded in the flowstone and dripwaters of Hodges Creek Cave, and the water extractable DOC from Adelaide Tarn (protein-like fluorescence was not observed in Dave's Cave, presumably owing to its higher elevation and cooler temperatures). Unfortunately, we do not have a method for calibrating DOM against a concentration standard and, therefore are unable to quantify the microbial contribution to the DOC pool.

Organic matter is processed along a continuum from intact plant matter to humic-like DOC, and microbial activity is a key component. However, from our dataset, we are unable to assess the impact of microbial activity on the DOC recorded in the flowstones. We have added some text to discuss and highlight this: *"The soil carbon cycle is extremely complex, encompassing the interactions between climate, hydrology, plants, microbial activity, and soils²⁷. Numerous temperature-sensitive ecosystem functions interact and respond to one another and may also display a non-linear response to chronic warming (e.g., vegetation productivity and carbon fixation, vegetation type, soil structure, microbial communities, and functioning)^{27,109,110}. Through the past 14,000 years, treeline elevations and vegetation density and diversity certainly responded to temperature changes^{62,91,111}, as did soil erosion rates⁶², and presumably microbial functions (as evidenced by fluctuations in protein-like fluorescence intensity at each study site).*

Although we measured humic-like DOC present in dripwaters and speleothems, the rate and extent to which DOC is processed and filtered during transport from soil to cave is poorly constrained⁴⁰. Changes in DOC properties may be explained by the soil continuum model¹², whereby organic matter (including SOC) is considered as 'a continuum of degrading compounds' ranging from intact plant material to highly oxidised carbon in carboxylic acids¹². In the context of speleothem science, DOC properties can be altered by microbial degradation during transport from soil to cave⁴⁰. In addition, although DOC concentrations are reliably recorded in the crystal lattice during precipitation as shown in laboratory studies³⁹, the effects of dynamic environmental conditions within a cave setting (e.g., pH, redox state, ventilation, microbial activity) are uncertain. Although we assessed the fluorescence intensity of the

protein-like fraction present in the flowstone cores, the extent to which microbial activity degraded DOC during transport and prior to calcite incorporation is unclear. Notably, a protein-like fluorescence signal (indicating microbial activity) was observed at Hodge Creek Cave, yet at Dave's Cave, a cooler cave at greater elevation, no protein-like fluorescence was observed, with the fluorescence signal dominated by humic-like DOC.

Reviewer 2, comment #5: Finally, would the authors consider adding multi-proxy data to support their new fluorescence proxy. Further LA-ICP-MS analyses of metal-transported trace elements, for example, would help understand the nature of the DOC being transported to the cave. Analysis of flowstone colour would be another proxy, I think one proposed previously by Pearson et al. (2020)? If colour calibrates with the fluorescence proxy, could it also enable a high-resolution record to be derived in a similar way that the authors use FTIR for the lake proxy record.

Response: As discussed in Pearson et al., 2020, speleothem colour can be a function of DOC incorporation. However, we chose to apply 3D EEM fluorescence analysis (to our knowledge, the first to use this approach to reconstruct DOC in speleothems) because this approach is reliable and highly sensitive for measuring DOC concentrations and deciphering between different fractions and sources of DOC. We agree with the reviewer that a method for using colour as a proxy for quantitative of DOC could be a valuable approach for future research (e.g., for higher-resolution analysis, or to allow non-destructive analysis of speleothem DOC).

Reviewer 3:

Reviewer 3, comment #1: The paper is presented on the understanding not only is there a temperature control on aqueous DOC dynamics, but there is also a rainfall control that can be clearly deciphered. However, this quickly begins to unravel as soon as the reader accesses the results section. I would suggest a minor re-structuring to redress these expectations for the reader.

Response: We thank the reviewer for this comment and have now included more hydroclimate proxies and undertook further investigation of Mg/Ca and Sr/Ca at our cave sites (please see 'Results and discussion' section). We agree that changes may have influenced DOC export but were likely subordinate to temperature overall. We have added some new paragraphs to the discussion, where we highlight the complexities of deciphering the influence of rainfall and hydrology on DOC export:

“Leaching of soil DOC requires rainfall volumes that exceed field capacity⁴⁰. However, the influence of rainfall amount on exported DOC loads and concentrations is complex, with several competing and complementary processes. For example, during drying events, higher DOC export can be driven by evapotranspirative concentration and increased microbial DOC-decomposition (compared to saturated conditions)^{102,103}, whilst soil drying and subsequent rewetting can drive increases in DOC release via priming of microbes, leading to sustained periods (i.e., years) of elevated DOC export¹⁰²⁻¹⁰⁴.

Notably the relationship between rainfall and DOC is inconsistent between the Hodge Creek and Dave's Cave archives. At Hodge Creek Cave, flowstone elevated Mg/Ca and Sr/Ca provide some evidence for drier conditions during the HCO, while high DOC is recorded at both Hodge Creek Cave and Adelaide Tarn during this period. However, at Dave's Cave, highly elevated DOC concentrations were recorded under drier conditions shortly after the onset of flowstone accumulation following glacial retreat from above the cave. Contrary to the trends recorded at Hodge Creek Cave, PCP declined alongside DOC at Dave's Cave, although DOC concentrations were still relatively high compared to contemporary values.

Although the temporal resolution of our samples is too low to allow for reconstructions of individual hydrological events, the occurrence of periods of drought at Hodge Creek and Adelaide Tarn during the HCO could be consistent with the elevated and oscillating DOC concentrations and higher prior-calcite precipitation reconstructed over this period. There are several mechanisms by which periodic drying events (associated with low DOC export from soil) and subsequent rewetting of soil may lead to increased DOC export. Such an effect could be plausibly explained by the 'enzymatic latch' mechanism¹⁰⁵, whereby decomposition is suppressed by the presence of phenolic compounds. Under excessively wet conditions, oxygen supply for decomposition is reduced, slowing DOC degradation¹⁰⁶,

but under dry conditions, a lower water table promotes oxic soil conditions, effectively eliminating phenolic compounds and their inhibitory effect on hydrolase enzymes^{107,108}. Drier soils also favour lower DOC solubility, enabling organic matter storage, which can be released as DOC upon rewetting, somewhat counterintuitively increasing export of DOC^{107,108}.

Numerous studies have demonstrated that aquatic DOC concentrations can remain elevated for years after rewetting following individual drying events^{105,107,108} because rewetting can stimulate the microbially-mediated breakdown of organic matter¹⁰⁴. Thus, drying events followed by rewetting and the 'enzymatic latch' mechanism may explain sustained elevated DOC concentrations during dry periods at Hodge Creek and Adelaide Tarn. However, DOC concentrations did not show any relationship with lower effective rainfall at Dave's Cave. Thus, while the impact of rainfall on DOC export is unclear, Holocene DOC concentrations were generally at their highest during the HCO at all three sites indicating a primary role for temperature driven processes in controlling long-term DOC export. We suggest this temperature DOC dependence reflects temperature-mediated changes in inputs (i.e. primary production) and degradation rates of soil C".

Reviewer 3, comment #2: The issue above is compounded through the sweeping use of Mg/Ca as a rainfall proxy with little consideration of the many alternative explanations that can be attributed to changing Mg/Ca in the speleothem record. The assumption seems to be (although the explanation for the reader is extremely poor), that prior calcite precipitation will alter the ratio of Mg to Ca, such that drier conditions raise the concentration of Mg relative to Ca. However, Mg/Ca ratios are impacted by many other factors. In the epikarst, concentrations of magnesium can be driven by incongruent dissolution, variable bedrock contact time (including contact with variable proportions of host dolomite), changing hydrological pathway, in addition to prior calcite precipitation. Within the cave, partitioning across the drip water-calcite interface can alter speleothem Mg concentrations via temperature, $p\text{CO}_2$, crystal fabric, and growth rate. Thus, interpretation of the Mg/Ca is often speleothem specific (let alone cave specific) and requires the support from other proxies to decipher the true nature of the Mg signal. This should include a thorough assessment of speleothem calcite $\delta^{18}\text{O}$, Sr concentration dynamics, co-variation between Mg and Sr, and other indicators (eg. Ba to eliminate growth rate effects) to thoroughly address the proxy signal interpreted from Mg/Ca in these two speleothems. There is plenty of literature available which addresses these drivers and competing effects. Unfortunately, these assumptions undermine all of the discussion regarding the effects of effective rainfall. It is thus little wonder that the data are difficult to interpret and seemingly competing effects between Mg/Ca and DOC are seen.

Response: We thank the reviewer for this helpful comment and agree that we gave a poor explanation and oversimplistic interpretation of the Mg/Ca proxy in our previous manuscript draft. We have now added a new section focused on Mg/Ca and Sr/Ca in the speleothem and dripwaters to provide support for the PCP control (see results and discussion section and Figure 2, or text pasted from the manuscript in response to Reviewer 1, comment #3. As highlighted in our responses to Reviewer 1, we also incorporated an extensive multi-proxy dataset to bolster the Mg/Ca data and interpretations, notably including Sr/Ca (for both flowstones) $\delta^{44}\text{Ca}$ (for Dave's Cave). We suggest that the data presented in Figure 3 show coherent trends in hydroclimate across the Holocene expressed in diverse proxies.

Reviewer 3, comment #3: The discussion surrounding some of the records is also a bit contradictory. There is confusion in the discussion of stage 2 regarding whether Dave's cave had higher, or lower, concentrations of DOC than the preceding stage. This is purely confusion in writing style and needs to be cleaned up. Further, as it is apparent the two speleothem records have two opposite trends between DOC and rainfall this does not make for a convincing conclusion regarding rainfall as a driver of DOC dynamics. Whilst I appreciate rainfall drivers of DOC are complex, the reader is not informed about this until later in the discussion. This leads to confusion and lack of confidence in the way the data has been interpreted. I suspect the use of Mg as a rainfall indicator has also led to much of this poor interpretation. It is also poor that drying trends (which incidentally are opposite to that expected) are identified post 6ka in the speleothem from Dave's cave, but most of this time period is covered by a hiatus (Supplementary information, Figure 6). Despite this invoked drying trend, the subsequent sentences then proceed to identify the hiatus as due to undersaturation of the drip waters in a climatically wet period. Confusing as currently written. Throughout, many of the correlations identified between records require statistical tests to prove these.

Response: We thank the reviewer for highlighting these stylistic issues. We have changed the writing style and removed references to 'stages'. We now refer to time-periods using their age values (see results and discussion section). We make it clear that DOC was at its highest concentration prior to the HCO but was still relatively high during the HCO (relative to other periods).

We agree that the relationship between rainfall and DOC export is unclear and complex. We have added several paragraphs discussing these complexities:

"Leaching of soil DOC requires rainfall volumes that exceed field capacity⁴⁰. However, the influence of rainfall amount on exported DOC loads and concentrations is complex, with several competing and complementary processes. For example, during drying events, higher DOC export can be driven by

evapotranspirative concentration and increased microbial DOC-decomposition (compared to saturated conditions)^{102,103}, whilst soil drying and subsequent rewetting can drive increases in DOC release via priming of microbes, leading to sustained periods (i.e., years) of elevated DOC export¹⁰²⁻¹⁰⁴.

Notably the relationship between rainfall and DOC is inconsistent between the Hodge Creek and Dave's Cave archives. At Hodge Creek Cave, flowstone elevated Mg/Ca and Sr/Ca provide some evidence for drier conditions during the HCO, while high DOC is recorded at both Hodge Creek Cave and Adelaide Tarn during this period. However, at Dave's Cave, highly elevated DOC concentrations were recorded under drier conditions shortly after the onset of flowstone accumulation following glacial retreat from above the cave. Contrary to the trends recorded at Hodge Creek Cave, PCP declined alongside DOC at Dave's Cave, although DOC concentrations were still relatively high compared to contemporary values.

Although the temporal resolution of our samples is too low to allow for reconstructions of individual hydrological events, the occurrence of periods of drought at Hodge Creek and Adelaide Tarn during the HCO could be consistent with the elevated and oscillating DOC concentrations and higher prior-calcite precipitation reconstructed over this period. There are several mechanisms by which periodic drying events (associated with low DOC export from soil) and subsequent rewetting of soil may lead to increased DOC export. Such an effect could be plausibly explained by the 'enzymatic latch' mechanism¹⁰⁵, whereby decomposition is suppressed by the presence of phenolic compounds. Under excessively wet conditions, oxygen supply for decomposition is reduced, slowing DOC degradation¹⁰⁶, but under dry conditions, a lower water table promotes oxic soil conditions, effectively eliminating phenolic compounds and their inhibitory effect on hydrolase enzymes^{107,108}. Drier soils also favour lower DOC solubility, enabling organic matter storage, which can be released as DOC upon rewetting, somewhat counterintuitively increasing export of DOC^{107,108}.

Numerous studies have demonstrated that aquatic DOC concentrations can remain elevated for years after rewetting following individual drying events^{105,107,108} because rewetting can stimulate the microbially-mediated breakdown of organic matter¹⁰⁴. Thus, drying events followed by rewetting and the 'enzymatic latch' mechanism may explain sustained elevated DOC concentrations during dry periods at Hodge Creek and Adelaide Tarn. However, DOC concentrations did not show any relationship with lower effective rainfall at Dave's Cave. Thus, while the impact of rainfall on DOC export is unclear, Holocene DOC concentrations were generally at their highest during the HCO at all three sites indicating a primary role for temperature driven processes in controlling long-term DOC export. We suggest this temperature DOC dependence reflects temperature-mediated changes in inputs (i.e. primary production) and degradation rates of soil C".

Reviewer 3, comment #3: I also felt as though there were many pieces of literature which were extremely relevant to the current manuscript, but which had not been paid attention. These include the many ground breaking studies by Baker et al. who pioneered the use of fluorescence EEM's to infer organic content of speleothem calcite; the more recent work by Blyth et al pioneering the extraction and molecular analysis of organic compounds from speleothem calcite, and the issues of contaminant molecules raised by Wynn and Brocks, 2014. (Rapid Communications in mass spectrometry). The work by Webb et al. 2014 (Journal of Quaternary Science) that addresses DOC in speleothem calcite from Australia and links this to the different proxy indicators of pluvial and arid phases; the wealth of literature surrounding interpretation of Mg and Sr in the speleothem record as indicators (or not) of prior calcite precipitation (Fairchild, numerous references; Sinclair et al., 2012. Chemical Geology) should also be addressed. There is also very little reference to the work of Worrall et al in the opening stages of the manuscript, paying reference to the work they have undertaken highlighting changing DOC fluxes from peatlands over the monitoring period.

Response: We thank the reviewer for pointing out these highly relevant manuscripts. We have now cited several of these seminal publications by Baker et al., Blyth et al., and Worrall et al., in the introduction section and throughout the manuscript. We have also cited Sinclair et al., 2012 in our interpretations of Mg/Ca and Sr/Ca.

In our methods section, we highlight the efforts undertaken to minimise the risks of sample contamination and cite Wynn and Brocks (2014): "Through the sample preparation process, caution was observed to minimise the risk of sample contamination¹⁷. For example..."

References from text pasted from manuscript:

- 1 Hellstrom, J., McCulloch, M. & Stone, J. A detailed 31,000-year record of climate and vegetation change, from the isotope geochemistry of two New Zealand speleothems. *Quaternary research* **50**, 167-178 (1998).
- 2 Hellstrom, J. & McCulloch, M. Multi-proxy constraints on the climatic significance of trace element records from a New Zealand speleothem. *Earth and Planetary Science Letters* **179**, 287-297 (2000).
- 3 McGlone, M. S., Hall, G. M. J. & Wilmshurst, J. M. Seasonality in the early Holocene: Extending fossil-based estimates with a forest ecosystem process model. *The Holocene* **21**, 517-526, doi:10.1177/0959683610385717 (2010).
- 4 McGlone, M. S., Hall, G. M. J. & Wilmshurst, J. M. Seasonality in the early Holocene: Extending fossil-based estimates with a forest ecosystem process model. *The Holocene* **21**, 517-526, doi:10.1177/0959683610385717 (2011).
- 5 Wilmshurst, J. M., McGlone, M. S., Leathwick, J. R. & Newnham, R. M. A pre-deforestation pollen-climate calibration model for New Zealand and quantitative temperature reconstructions for the past 18 000 years BP. *Journal of Quaternary Science* **22**, 535-547 (2007).

- 6 McGlone, M. S., Turney, C. S. M., Wilmshurst, J. M., Renwick, J. & Pahnke, K. Divergent trends in land and ocean temperature in the Southern Ocean over the past 18,000 years. *Nature Geoscience* **3**, 622-626, doi:10.1038/ngeo931 (2010).
- 7 Wang, M. *et al.* Global soil profiles indicate depth-dependent soil carbon losses under a warmer climate. *Nature Communications* **13**, 5514, doi:10.1038/s41467-022-33278-w (2022).

REVIEWERS' COMMENTS

Reviewer #1 (Remarks to the Author):

Dear authors, many thanks for providing a detailed description and revision of your manuscript. I found that all my points have been clearly addressed, resulting in a much stronger and clearer manuscript.

I have no further comments and I recommend the manuscript for publication by Nature Communications. I think this is a very nice and novel study that would fit well within the scope of the journal.

Reviewer #3 (Remarks to the Author):

This is a re-review of paper NCOMMS-23-30654A.

I have considered in detail the response of the authors to my specific reviewer comments (Reviewer 3). I consider these to be well addressed with extensive and appropriate changes made to the manuscript accordingly. It is pleasing to see the better inclusion of hydroclimate proxies and a more robust interpretation of the way in which rainfall dynamics may drive DOC release. There has also been significant correction to the way in which Mg/Ca ratios have been used. Many thanks also for paying better attention to the many seminal papers in the published domain that set the stage for the work undertaken in this publication.

I have only briefly addressed the comments made by the other reviewers and cannot comment extensively on these. However, I can qualify that all three reviewers commented on the poor use of the Mg/Ca proxy, and that this has now been addressed accordingly.

I would now consider this paper worthy of publication.